# Comparative genome analysis provides a foundation for defining salvinorin A biosynthesis in *Salvia divinorum*

Haixiu Li [1,9], Yuwei Sun[2,9], Wenliang Xu[3], Baocheng Sun [3], Song Wu [4], Chao Li [1,5,6], Junwei Zhao[7], Cathie Martin [8] ✉ & Evangelos C. Tatsis [3] ✉

The synthesis of salvinorin A by *Salvia divinorum* is of considerable interest for developing pain relief, anti-opioid addiction and antidepressant medications, but progress has been compromised by limited access to plant material and the complexity of the biosynthetic pathway. Initially using *S. splendens*, a closely related species, the first steps in the biosynthetic pathway have been elucidated. Here, by preparing a genome sequence for *S. divinorum*, we are able to undertake comparative genomic analyses with closely related species that do not produce salvinorin A. We establish the genetic basis for additional activities in salvinorin A biosynthesis involving cytochrome P450 and methyltransferase enzymes. Our genome-based, microevolutionary approach provides insight into how specialized furanoclerodanes of immense pharmacological importance evolved. These results lay a clear path for identification of the remaining steps in the biosynthetic pathway which would allow synthetic production for the development of new therapeutics.

*Salvia divinorum* Epling and Játiva, is a flowering plant from the family Lamiaceae, native to the cloud forests of mountain ranges in the state of Oaxaca in southwestern Mexico (Fig. 1). This species has attracted interest for the transient intense hallucinogenic effects of leaf infusions. *S. divinorum* holds a strong cultural significance for indigenous Mazatecan tribes who use fresh leaves in Shamanic ceremonies for divination[1–3]. While large doses of *S. divinorum* infusions cause hallucination and can alter consciousness, smaller doses are used as a tonic or panacea[1] for pain relief and to treat inflammatory conditions, arthritis, rheumatism, as well as substance abuse[1,3,4]. No wild populations have been identified, and its distribution is anthropogenic based on vegetative propagation by the Mazatecs. *S. divinorum* flowers infrequently and exhibits reduced pollen fertility leading to minimal seed yield (3% in greenhouses with cross-pollination)[3].

The hallucinogenic activity of *S. divinorum* has been linked to the presence of a clerodane diterpenoid salvinorin A[2,5]. Salvinorin A has been identified by the National Institute of Mental Health (NIMH) Psychoactive Drug Screening Program as a promising candidate for treating mental disorders. Salvinorin A stimulates the κ-opioid receptor, demonstrating strong binding efficiency and high selectivity compared to other opioid receptors[6–9]. Despite its functional similarity to natural hallucinogens such as psilocybin and mescaline, salvinorin A is structurally unique among known κ-opioid receptor agonists due to its lack of a basic nitrogen atom[7]. The acute activation of κ-opioid receptors by salvinorin A can reduce addiction to recreational drugs, suggesting a potential use in rehabilitation therapies[10–13]. Both salvinorin A and its synthetic derivatives have advanced to pre-clinical and phase 1 clinical trials, aiming to develop new treatments for mental disorders, drug addiction, and pain relief[12–14]. Other clerodanes from

[1]School of Life Sciences, East China Normal University, Shanghai, China. [2]The SATCM Key Laboratory for New Resources & Quality Evaluation of Chinese Medicine, Institute of Chinese Materia Medica, Shanghai University of Traditional Chinese Medicine, Shanghai, China. [3]School of Breeding and Multiplication (Sanya Institute of Breeding and Multiplication), Hainan University, Sanya, China. [4]Max Planck Institute for Chemical Ecology, Jena, Germany. [5]Institute of Advanced Agricultural Science and Technology, East China Normal University, Shanghai, China. [6]Institute of Eco-Chongming, East China Normal University, Shanghai, China. [7]Beijing Life Science Academy, Beijing, China. [8]The John Innes Centre, Norwich, United Kingdom. [9]These authors contributed equally: Haixiu Li, Yuwei Sun. ✉e-mail: cathie.martin@jic.ac.uk; evangelos.tatsis@hainanu.edu.cn

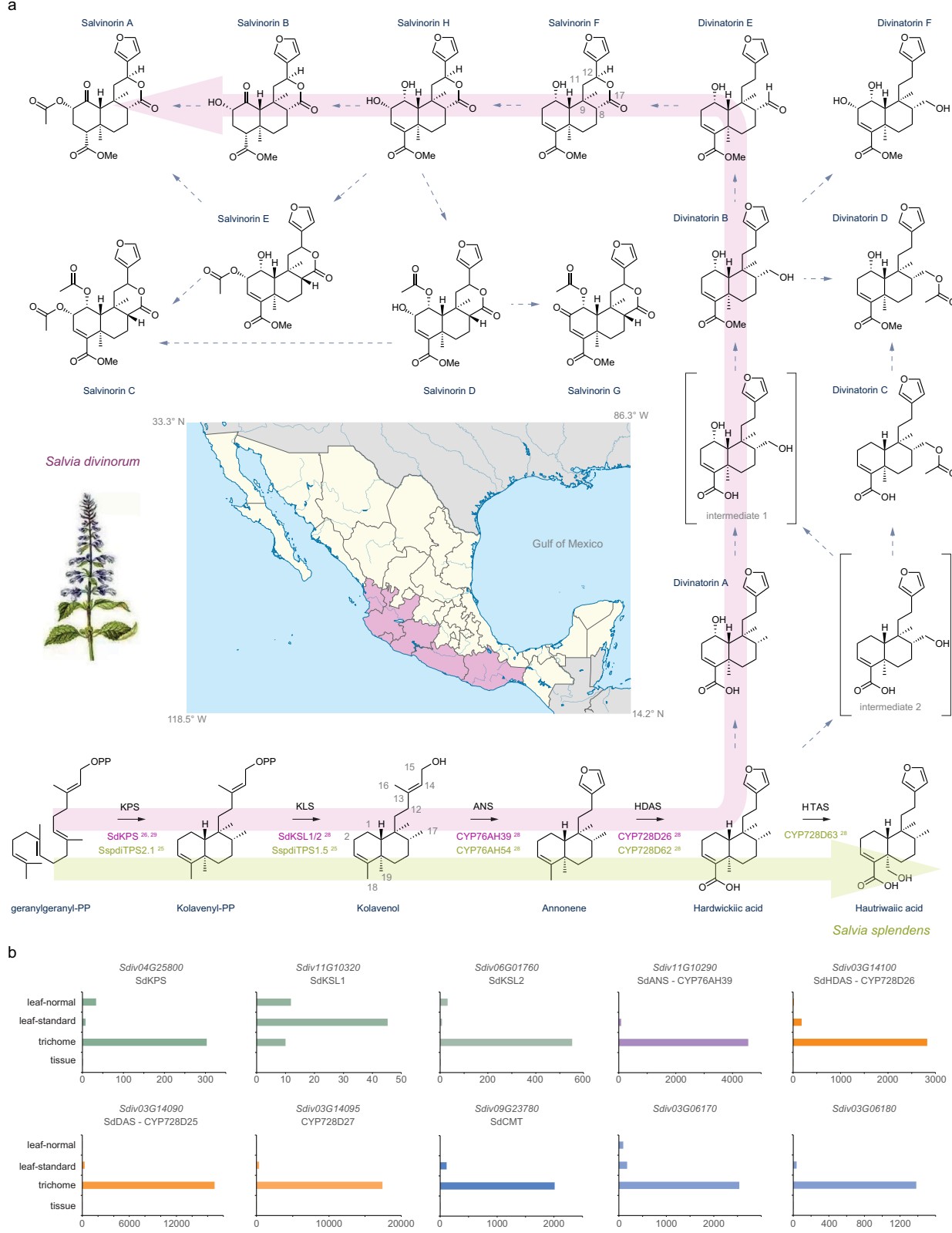

a

b

*S. divinorum*, such as kolavenol, hardwickiic acid, and salvinorin B, also show biological activities[14,15], indicating the potential of furanoclerodanes for new therapeutics that target central nervous system pathologies, including relief from neurological pain and visceral hypersensitivity[15,16]. Clinical research on salvinorin A and other furanoclerodanes from *S. divinorum* is limited currently due, in part, to the lack of established cultivation methods for *S. divinorum*[17]. While total chemical synthesis of salvinorin A is possible[18–20], it is not commercially viable. Elucidating the biosynthetic pathway of salvinorin A would facilitate its production and enable structural diversification through synthetic biology, thereby facilitating research into its potential medical applications.

*S. divinorum* belongs to the subgenus Calosphace, which is the most diverse of the 11 subgenera of *Salvia*, with 587 species. The

**Fig. 1 | Geographical distribution of *S. divinorum* in southwest Mexico and the proposed biosynthetic pathway to salvinorin A. a** The enzymatic steps involved in the biosynthesis of the furanoclerodane precursor hardwickiic acid are highlighted in purple for *S. divinorum* and in green for *S. splendens*[25,26,28,29], while the uncharacterized steps are indicated by dashed arrows. The purple area in the map indicates the distribution of *S. divinorum* in southwest Mexico. The depicted Mexico map is modified (painted purple) from "Creative Commons Mexico States blank map.svg" created by Yavidaxiu, Sémhur, and Kmusser, used under CC BY-SA 4.0. Map center latitude: 23° 36' 00" N; longitude: 102° 30' 00" W. **b** Expression profile of genes in *S. divinorum* encoding clerodane biosynthetic enzymes SdKPS, SdKSL1, SdKSL2, SdANS, SdHDAS, cytochrome P450 enzymes CYP728D25 (SdDAS), CYP728D27, and SAM methyl transferases (*Sdiv09G23780*, *Sdiv03G06170*, and *Sdiv03G06180*) which are candidates for catalyzing the methylation of the C-18 carboxy functional group. Data were obtained from mapping EST data to the genome data presented in this manuscript. Source data are provided as a Source Data file.

subgenus, Calosphace, consists of species endemic to the Americas, with most distributed in the Neotropics, with a center of diversity in Mexico, extending to the Andes, the Antillean islands and Brazil as endemic centers[21–23]. European and Asian genera of *Salvia* are rich in labdane and abietane diterpenoids such as carnosic acid, forskolin, and tanshinones[24]. Neotropical *Salvia* species such as *S. divinorum*, *S. splendens*, and *S. hispanica* are rich in clerodane diterpenoids[21,22]. While clerodanes are present in other genera of the family Lamiaceae, phylogenomic and biochemical research has shown that clerodane biosynthesis is polyphyletic in origin[25].

Salvinorin A is a furanoclerodane synthesized and accumulated in trichomes[26] of *S. divinorum,* as occurs with other clerodanes in the family Lamiaceae, such as scutebarbatine A in *Scutellaria barbata*[27] and salviarin in *S. splendens*[28]. Clerodane biosynthesis in *S. divinorum* starts in plastids, where geranylgeranyl diphosphate (GGPP) from the methyl erythritol pathway (MEP) is subject to cyclization and two methyl migrations by a class II diterpene synthase, kolavenyl diphosphate synthase (SdKPS)[26,29] followed by the lysis of the phosphate bond and formation of kolavenol by class I diterpene synthases with kolavenol synthase activity (SdKSL1/2)[28]. It is not clear if diterpenes are actively exported from the plastids[30] or transferred via hemifusion at contact sites between plastid and ER[31,32] for later oxidations by cytochrome P450 enzymes. In *S. divinorum*, CYP76AH39[33] (named annonene synthase (SdANS)) catalyzes the furan ring formation in annonene[28], followed by oxidation of the methyl group of C-19 to carboxylic acid by CYP728D26 (named hardwickiic acid synthase (HDAS)) to form hardwickiic acid[28] (Fig. 1). However, the catalytic steps involved in the biosynthesis of salvinorin A after the formation of hardwickiic acid remain largely uncharacterized. Based on clerodanes isolated from *S. divinorum*, a biosynthetic pathway to salvinorin A and other salvinorins via divinatorins has been proposed[26,34,35], involving the formation of divinatorin intermediates by oxidations of C-1, C-17 and methylation of the free carboxyl group at C-18 of the clerodane scaffold (Fig. 1a).

Whole-genome sequencing is an effective strategy for identifying genes encoding enzymes active in metabolic pathways for the biosynthesis of specialized metabolites and for tracing their evolutionary origins, as exemplified in the family Lamiaceae[25,36–39]. The genome sequences of several members of the genus, *Salvia*, including two species from the subgenus Calosphace, *S. hispanica*, and *S. splendens*, together with other species from the Asian lineage, *S. miltiorrhiza* and *S. bolwleyana*, and from the European lineage, *S. rosmarinus*, and *S. officinalis*, enable comparative analyses to facilitate understanding of the production of salvinorin A and furanoclerodanes in *S. divinorum*.

Here, we report a genome sequence of *S. divinorum* at the chromosome level using a combination of Illumina and PacBio data, which is assembled using information from HiC technologies. Comparative genomic analysis is performed using published genomes of *Salvia* species *S. hispanica*, S. *splendens*, *S. miltiorrhiza*, *S. bolwleyana, S. rosmarinus*, and *S. officinalis* with *Sesamum indicum* as an outgroup. The evolutionary path for the biosynthesis of salvinorin A in *S. divinorum* appears to have arisen following tandem gene duplication in *S. divinorum* and specific recruitment of genes encoding CYP728D subfamily proteins, which oxidize hardwickiic acid to form the divinatorins. A carboxy methyltransferase (SdCMT) proposed to be responsible for the methylation of the C-18 carboxyl group is confirmed by screening the genome and transcriptome sequences and undertaking enzyme assays in *Nicotiana benthamiana*.

## Results

### *S. divinorum* genome sequencing, assembly, and annotation

The DNA for genome sequencing of *S. divinorum* was extracted from leaves of a single plant maintained in the greenhouse. Using flow cytometry, we determined the genome size to be 536 Mb (Supplementary Table 1 and Supplementary Fig. 1), while k-mer analysis and GenomeScope[40] estimated the genome size to be 550 Mb (Supplementary Table 2 and Supplementary Figs. 2 and 3). We obtained 36.77 Gb (68×) sequence data by PacBio, 131.80 Gb (246×) sequence data by HiC, and 20 Gb (37×) sequence data by Illumina. The PacBio long-read sequencing data were used for genome assembly. By using the Hifiasm[41] and Purge_Dups[42] pipelines, the nonredundant genome sequences yielded an assembled size of 558.96-Mb, with a total number of 215 contigs and an N50 value of 36.11 Mb (Supplementary Table 3). A HiC (in vivo fixation of chromosomes) library was then employed to anchor the nonredundant sequences onto 11 pseudochromosomes and provide a refined first version of the genome using Juicer tools and 3D-DNA pipelines[43] (Supplementary Table 3 and Supplementary Fig. 4). Finally, 541.87 Mb of the nonredundant sequence, accounting for 96.94%, were anchored onto 20 super-scaffolds with a super-scaffold N50 of 52.13-Mb, and a GC content of 37.34% (Supplementary Table 3). All the super-scaffolds could be placed in one of eleven groups. We ran smudgeplot[40] to assess the ploidy, and the result showed that *S. divinorum* is a diploid (Supplementary Fig. 5), consistent with earlier reports[44]. The number of groups, hereafter referred to as pseudochromosomes, corresponded to the reported number of chromosomes (1n = 11, 2n = 22), representing a high-quality chromosome-level monoploid genome assembly which bridged the gaps in the assembly of the *S. divinorum* genome reported previously[45] (Supplementary Figs. 6–8).

BUSCO (Benchmarking Universal Single-Copy Orthologs) evaluation of the gene models in the genome assembly showed 98.8% completeness with queries against the embryophyta_odb10 database[46] (Supplementary Table 4). The pseudochromosome length for *S. divinorum* ranged from 29.06 Mb (Chr10) to 72.59 Mb (Chr03) (Supplementary Table 5). LTRs (Long terminal repeats of retroelements) were the most abundant repeat sequences of the genome, comprising 55.94%, followed by DNA transposable elements at 3.18% (Supplementary Table 6). The non-coding RNAs in the *S. divinorum* genome included 94 miRNAs, 1880 tRNAs, 8862 rRNAs, and 653 snRNAs (Supplementary Table 7). An overview of the genome assembly, including gene density, non-coding RNA densities and repeats, is shown in Fig. 2. A total of 41,531 genes were annotated functionally by querying publicly available databases, including MAKER (v3.01.03), SwissProt, NR, Pfam, TrEMBL, KEGG, and InterProscan. The top match from each database was linked to the protein-coding gene. These statistics show that we obtained a chromosome level genome assembly with high completeness compared with other sequenced genomes from the genus *Salvia* (Supplementary Fig. 9) (*S. splendens*[47], *S. hispanica*[48], *S. miltiorrhiza*[49], *S. bowleyana*[50], *S. rosmarinus*[51], and *S. officinalis*[52]) as well as the earlier report for *S. divinorum*[45], whose quality and completeness were not independently assessed in this study.

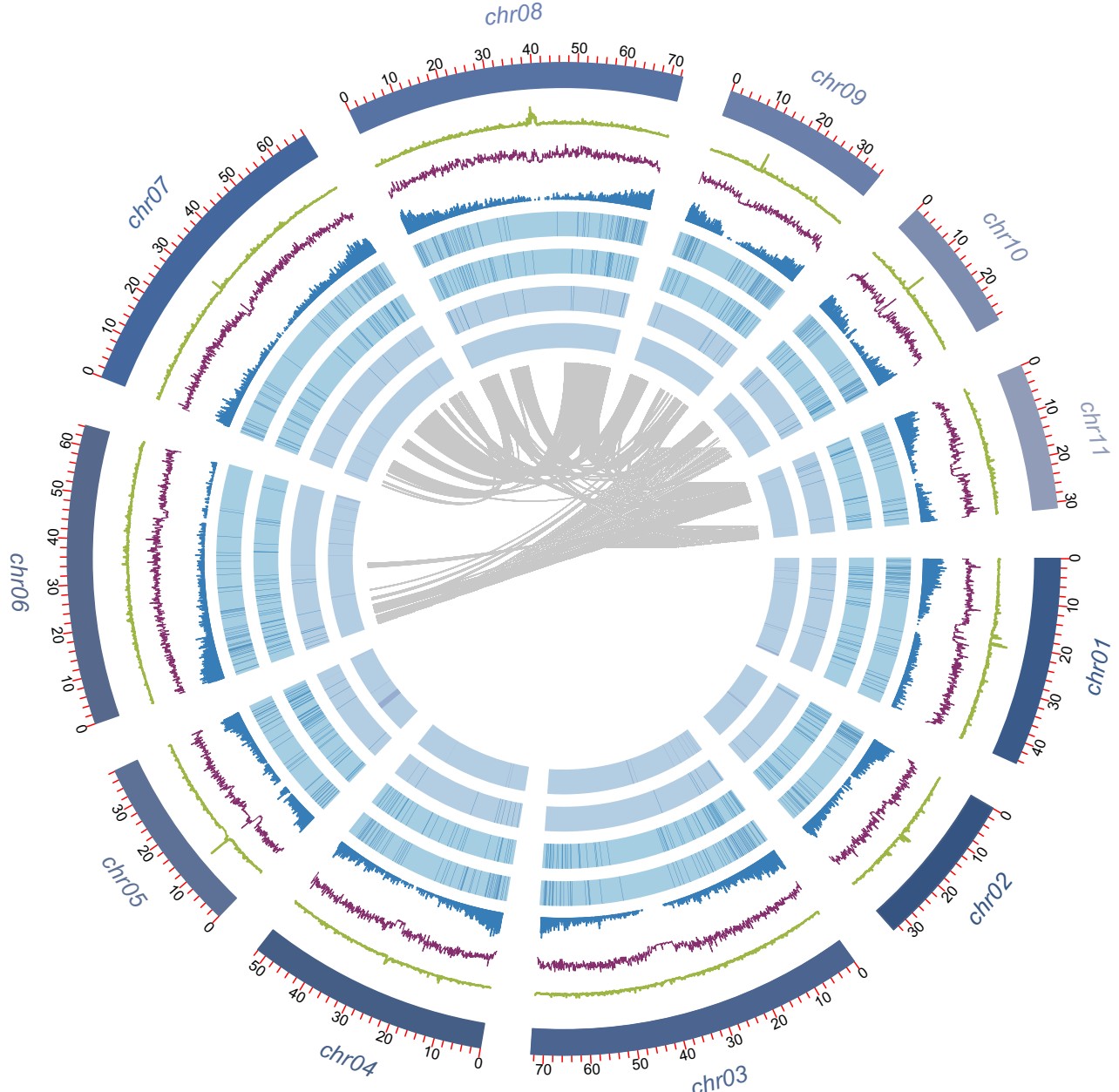

**Fig. 2 | Overview of the *S. divinorum* genome assembly.** The CIRCOS plot depicts the features of the *S. divinorum* genome assembly (size 541.87 Mb). From inner to outer layers: the gray links represent intraspecies syntenic relationships, and the layers show, sequentially, the densities of simple sequence repeats, rRNA, miRNA, snRNA, tRNA, and protein-coding genes. The purple layer shows the frequency of transposable elements (TE), and the green layer shows the repeat density in a 100-kb sliding window. In the outer layer, solid, blue bars represent individual chromosomes. Source data are provided as a Source Data file.

## Comparative genomic analysis

We compared our genome assembly with the recently published *S. divinorum* genome; our version consisted of 11 pseudochromosomes reflecting the reported number of haploid chromosomes and was more continuous and contained a higher number of genes than the published version[45] (Supplementary Figs. 6–8). To estimate the evolutionary origins of *S. divinorum*, we compared our genome assembly for *S. divinorum* with ten other genome assemblies; two American *Salvia* species (*S. splendens*, *S. hispanica*), two Asian *Salvia* species (*S. miltiorhiza*, *S. bowelyana*), two European *Salvia* species (*S. rosmarinus*, *S. officinalis*), two species from the order, Lamiales (*Sesamun indicum*, *Andrographis paniculata*), together with a euasterid and a eurosid species (*Solanum lycopersicum* and *Arabidopsis thaliana*, respectively) as outgroups. Based on analysis of gene family clustering, we identified 27,924 gene families, of which 8032 were shared by all 11 species. In total, 256 gene families were shared by all seven species in the genus *Salvia*, while the two Asian *Salvia* species shared 636 gene families and the two European *Salvia* species shared 167 gene families. Using all the identified orthogroups[53] among the selected 11 species (Fig. 3), a phylogenetic tree of species was constructed. The branching order in the tree showed that European *Salvia* species diverged from Asian and American subgenera 23.3 Million Years Ago (MYA) (Fig. 3a), followed by divergence between Asian and American *Salvia* species 21.6 MYA (Fig. 3a). As expected, *S. divinorum* is related closely to the two other American *Salvia* species in the subgenus, Calosphace, *S. splendens* and *S. hispanica*, which diverged from *S. divinorum* around 10.6 MYA

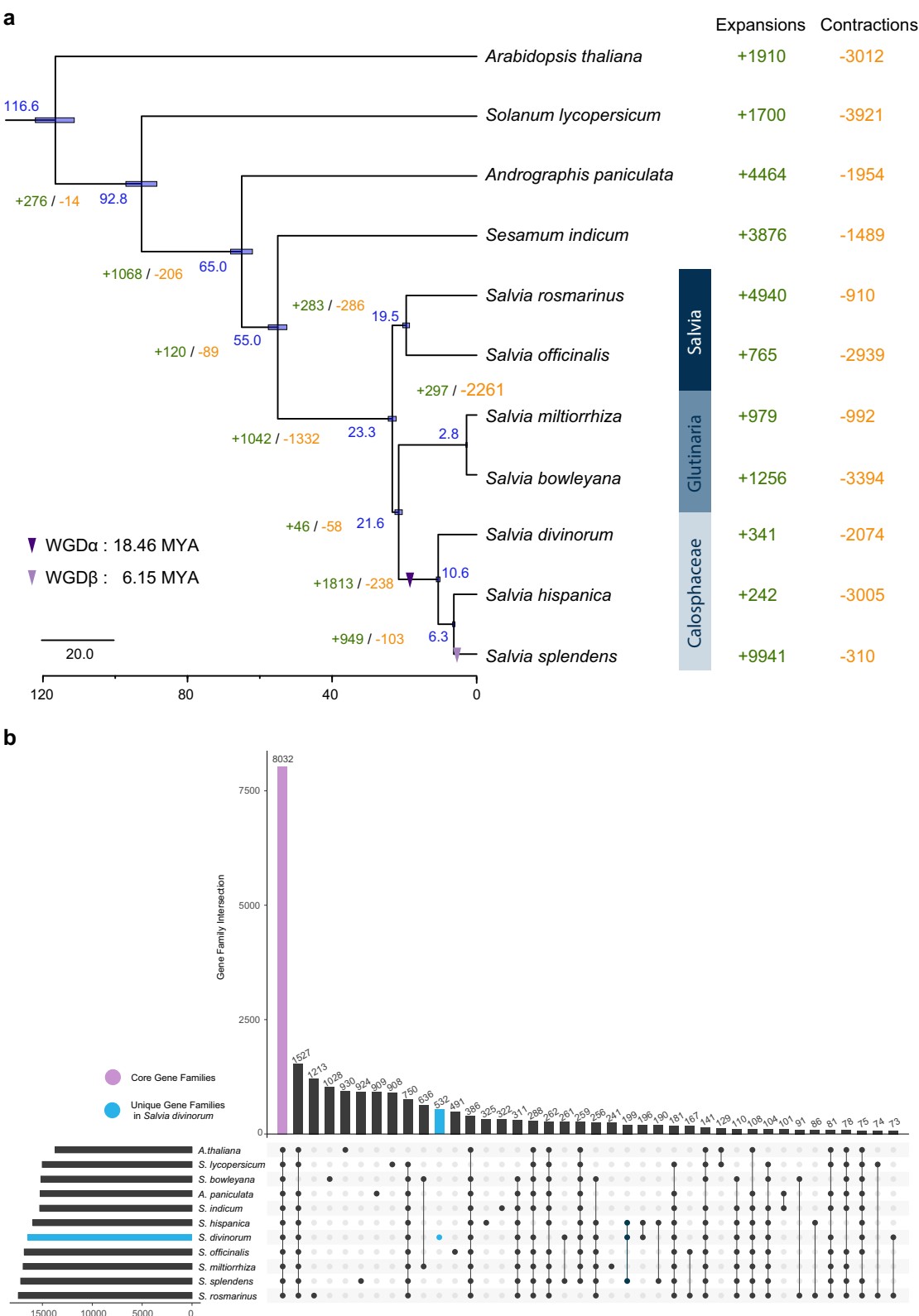

**Fig. 3 | Comparative genomic analysis. a** Phylogenetic analysis and divergence time estimations among 11 plant species. The tree was constructed using all the identified orthogroups among the selected 11 species. Divergence times (MYA) are indicated by the blue numbers beside each branch node. The number of gene-family contraction and expansion events is shown by the yellow and green numbers beside each node, respectively. The purple triangles indicate the recent WGD events in the three *Salvia* species of the subgenus, Calosphace. In blue frames are highlighted the different *Salvia* subgenera. **b** Shared and unique gene families between the eleven species. The Upset plot shows the core and unique gene families across the eleven species. Intersections of overlapping gene families have been represented by linked dots, and the number of intersections are shown in the bar plot. Source data are provided as a Source Data file.

(Fig. 3a). The three American *Salvia* species shared 199 gene families compared with their most recent common ancestor (MRCA) (Fig. 3b), and in *S. divinorum*, 341 gene families have expanded and 2074 contracted in total (Fig. 3a).

Intragenomic and self-alignment analysis between and within the pseudochromosomes revealed paralogous relationships among the 11 *S. divinorum* pseudochromosomes with 331 major duplications (Fig. 2). Collectively, *S. divinorum* has 9526 pairs of paralogous genes. Based on the density distribution of the synonymous substitution rate per gene ($Ks$) between the collinear paralogous genes, a $Ks$ peak at 0.24 was calculated, indicating a major duplication event (Supplementary Fig. 10). Similar analyses of the genome assemblies of *S. hispanica* and *S. splendens*, showed that all the three American *Salvia* species share this major duplication event, most likely a whole-genome duplication (WGD), which occurred after the divergence from the Asian *Salvia* species (Fig. 3a and Supplementary Fig. 8) estimating the time of WGDα around 18.46 MYA. A $Ks$ peak at 0.08 for *S. splendens* (Supplementary Fig. 10) indicated that a second WGD (WGDβ), specific to the *S. splendens* lineage, occurred 6.15 MYA, just after the split between *S. splendens* and *S. hispanica*. Our estimates of the timing of WGDα in *S. divinorum* and WGDβ are consistent with what has been proposed for *S. splendens*[47] and *S. hispanica*[48]. Aligning the *S. divinorum* genome assembly with *S. hispanica* and *S. splendens* (Supplementary Figs. 11–13) revealed 492 syntenic partnering blocks between *S. divinorum* and *S. hispanica* and 972 syntenic partnering blocks between *S. divinorum* and *S. splendens*. Additionally, there were 26,004 syntenic gene pairs between *S. divinorum* and *S. hispanica* (Supplementary Fig. 12) and 41,264 gene pairs between *S. divinorum* and *S. splendens* (Supplementary Fig. 13). The doubled number of syntenic partnering blocks and genes between *S. divinorum* and *S. splendens* is associated with WGDβ, specific to the *S. splendens* lineage.

## Mapping the characterized genes encoding enzymes in salvinorin A biosynthesis onto the *S. divinorum* genome

We used three different sets of RNA-seq data from NCBI, described as plant (SRR15247010)[21], leaf trichome (SRR3716680)[29], standard leaf (SRR3746809)[33,54], and normalized leaf (SRR3746810)[33,54]. The RNA-seq were mapped and aligned to the genome of *S. divinorum*. We then investigated the expression of characterized genes across four different samples. Previous reports have shown that salvinorin A accumulates in glandular trichomes[55,56] and this information has been used to identify the first step in the biosynthesis of salvinorin A performed by a class II diterpene synthase with kolavenyl diphosphate synthase activity (SdKPS)[26,29]. SdKPS is encoded by gene *Sdiv04G25800* on chromosome 4 with very high expression in trichomes (Fig. 1b). Two class I diterpene synthases, *SdKSL1* and *SdKSL2*, are expressed in trichomes and transform kolavenyl diphosphate to kolavenol[28]. SdKSL1 is encoded by the gene *Sdiv11G10320* on chromosome 11, and SdKSL2 is encoded by the gene *Sdiv06G01760* on chromosome 6. While *SdKSL1* is expressed broadly across different plant tissues, including trichomes, *SdKSL2* is highly expressed in trichomes and has a similar expression profile to other genes operating in the pathway (Fig. 1b).

Despite an early assumption of a monophyletic origin of clerodane biosynthesis, which occurs widely in species of the family Lamiaceae, biochemical and phylogenomic data have indicated distinct evolutionary lineages, especially in the genera *Scutellaria* and *Salvia*, suggesting polyphyletic origins of clerodane diterpenoid biosynthesis in the family Lamiaceae. More specifically, the evolution of class II clerodane synthases from a copy of a gene encoding *ent*-copalyl diphosphate synthase, involved in gibberellin metabolism, is the result of an ancient whole-genome duplication (WGD) which occurred ~66 MYA, followed by neofunctionalization under repeated evolution. The polyphyletic nature of clerodane biosynthesis in the Lamiaceae is supported by the phylogenetic distances between class II clerodane synthases from Scutellaria and *Salvia*[44], as established by both classical

and Bayesian phylogenies and the absence of identical enzymatic activities between syntenic genes in comparisons of some, closely related species. SdKPS is phylogenetically close to the *S. splendens* class II clerodane synthase SsdiTPS2.1[25], and syntenic analysis (Fig. 4) demonstrated that SdKPS lies in the same syntenic region as genes encoding other class II clerodane synthases in species of the family Lamiaceae. Syntenic analyses of class I clerodane synthases in the genus *Salvia*, showed that *SdKSL2* is syntenic to the kolavenol synthase gene from *S. splendens* SspdiTPS1.5 (Fig. 4). Class I clerodane synthases making kolavenol were recruited from genes in abietane metabolic pathways, specifically miltiradiene synthase, but show different/independent genomic and phylogenetic origins in *Salvia* and *Scutellaria*. Syntenic analysis showed that these *KLS* genes are specific to the genus *Salvia* (Fig. 4), and their absence in other lineages of species of the Lamiaceae, such as *Scutellaria* or *Callicarpa*, indicates evolutionary events specific to the genus *Salvia* (Fig. 4). While class II clerodane synthases—like SdKPS—can catalyze the initial cyclization and methyl migrations of GGPP to form kolavenyl diphosphate, class I activity is essential for subsequent phosphate lysis and formation of kolavenol, a critical precursor for downstream oxidations leading to furanoclerodanes like salvinorin A. The absence of conserved syntenic ortholog class I clerodane synthases genes in Scutellaria[25] and Callicarpa species, is in full agreement with the conclusion that there has been rapid evolution of clerodane biosynthetic pathways by convergence and that the repeated evolution a of class II diterpene synthase activity alone is insufficient for clerodane diterpene formation in the Lamiaceae.

## The emergence of annonene synthase is linked to a WGD event shared between species of the subgenus Calosphace

The first step in biosynthesis of salvinorin A after the formation of the clerodane scaffold in plastids involves the conversion of kolavenol to annonene by annonene synthase, SdANS, a cytochrome P450 from the CYP76AH subfamily (CYP76AH39)[28]. SdANS is encoded by the gene *Sdiv11G10290* on chromosome 11, and the highest expression is observed in leaf trichomes (Fig. 1b). Phylogenetic analyses (Fig. 5a) showed that the activity of annonene synthase, in both *S. divinorum* and *S. splendens*, evolved by recruitment of genes encoding cytochrome P450 enzymes originally with ferruginol synthase activity and neofunctionalization towards annonene synthase activity[25,28], although it is unclear whether neofunctionalization was a result of a tandem duplication or some other duplication event. We performed phylogenomic analysis, aligning the syntenic blocks of the seven *Salvia* species with the phylogenetic tree of the species. The gene encoding ferruginol synthase lies in a highly conserved biosynthetic gene cluster (BGC) consisting of genes encoding class II (copalyl diphosphate synthase), class I (miltiradiene synthase) diterpene synthases and cytochrome P450 enzymes, mainly belonging to the CYP76AH subfamily, with ferruginol synthase activity. Since this cluster of metabolic genes is highly specific to the family Lamiaceae[25,57] and this genomic area is highly enriched in tandem gene duplications, we avoided using a member of the Lamiaceae as an outgroup and opted to use *Sesamum indicum* as the outgroup for the phylogeny.

The phylogenetic tree (Fig. 5a) showed that the genes encoding annonene synthase in both *S. divinorum* and *S. splendens* (SdANS, SspANS) align in the same clade (within the purple box in Fig. 5) as the three enzymes from *S. hispanica* and one other from *S. splendens*. Phylogenomic analysis (Fig. 5b) show that the Asian species of *Salvia, S. miltiorhiza, S. bowleyana* together with the European species, *S. officinalis*, contain one syntenic genomic block containing genes encoding enzymes of CYP76AH subfamily, whereas the three American *Salvia* species showed duplicated copies of this genomic area. Since our comparative genomic analysis indicated that these three species shared a common WGD event which occurred ~18.6 MYA, after the divergence of the American and Asian *Salvia* clades ~21.6 MYA (Fig. 5b). The effect of this WGD event (WGDα) at the genomic level is depicted

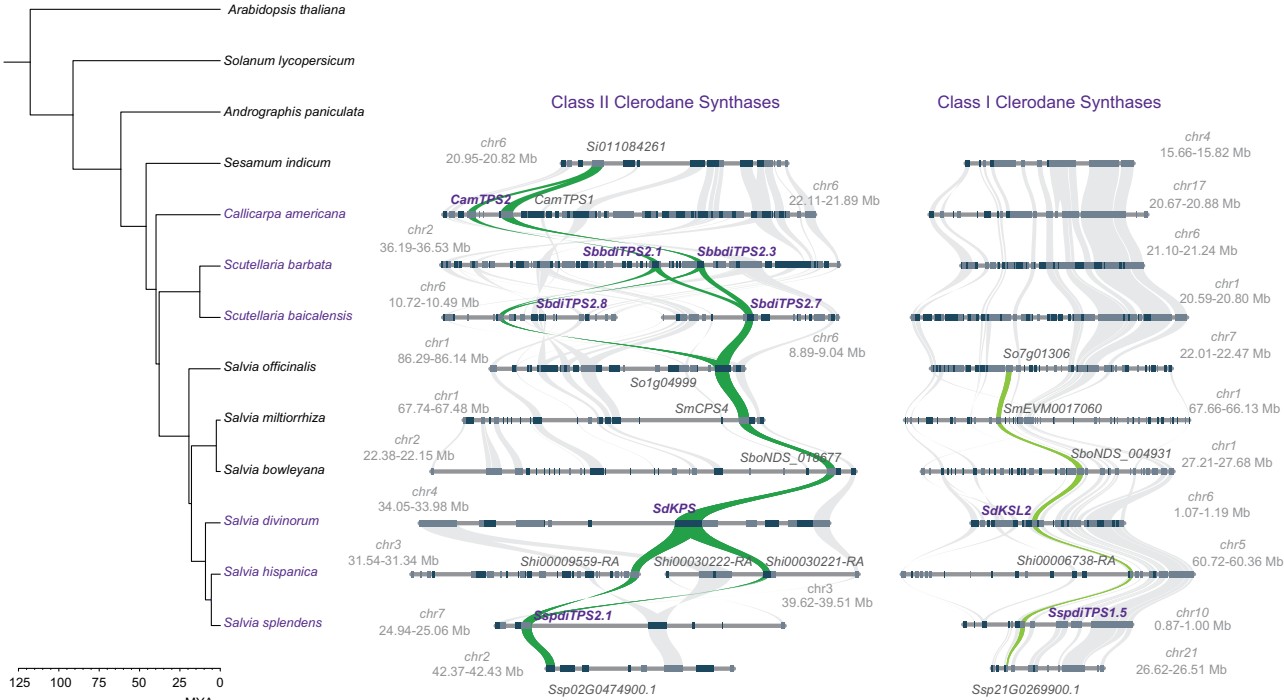

**Fig. 4 | Phylogenomic diagram aligning the phylogenetic tree and syntenic analyses of the class II and class I diterpene synthases involved in clerodane biosynthesis in the family Lamiaceae.** The species producing clerodanes are highlighted in purple fonts. The genes shown in purple correspond to functionally characterized enzymes with class II and class I clerodane synthase activities. The syntenic relationships between class II and class I diterpene synthases are high-lighted by the green ribbons.

in the synteny analysis by a purple ribbon (Fig. 5b). The cytochrome P450 genes lie in a genomic fragment (highlighted by the gold ribbon in Fig. 5b) and retain their ferruginol synthase activity[25,28]. Almost all the genes contained in the duplicated (second) genomic fragment (highlighted by the purple ribbon, Fig. 5b) belong to the same phylogenetic clade as SdANS and SspANS and we hypothesize that the WGDa provided the genes for the emergence of annonene synthase activity. Since the cytochrome P450 enzymes encoded by the CYP76AH36 gene(s) show some annonene synthase activity while the enzymes encoded by the CYP76AH87 genes do not[28], it can be hypothesized that SdANS and SspANS evolved from a copy of a CYP76AH36 ancestral gene. The neofunctionalization of CYP76AH enzymes in American *Salvia* species—from ancestral ferruginol synthase to anno-nene synthase activity - was likely facilitated by the whole-genome duplication event (WGDα) shared by these neotropical *Salvia* species. Following duplication, one copy may have retained the original fer-ruginol synthase function, associated with abietane diterpenoid bio-synthesis, while the other acquired annonene synthase activity, enabling the formation of the furan ring in clerodane precursors such as those leading to salvinorin A[28]. This divergence underscores how WGD events can drive the evolution of specialized metabolic pathways in plants.

## Enzymes of the CYP728D subfamily play a key role in the expansion of clerodane biosynthesis in neotropical *Salvia* species

In both *S. divinorum* and *S. splendens*, cytochrome P450 enzymes from the CYP728D subfamily, CYP728D26 and CYP728D62, respectively, oxidize the methyl C-18 of annonene to carboxylic acid to form hard-wickiic acid (Fig. 1)[28]. CYP728D63 from *S. splendens* then oxidizes the C-19 of hardwickiic acid to form hautriwaic acid[28]. In *S. divinorum*, a gene encoding CYP728D26 oxidizes the methyl group on the C-18 of annonene to carboxylic acid to form hardwickiic acid (Fig. 1)[28], but there is no reported oxidation of C-19 to form hautriwaic acid (Fig. 1).

Examination of the genome sequence of *S. divinorum* in the region of the gene encoding CYP728D26 (SdHDAS) revealed three additional genes encoding CYP728D proteins (Fig. 6a), which appeared to have arisen following tandem duplication. Of these, two encoding CYP728D25 and CYP728D27, appeared to be highly expressed in tri-chomes of *S. divinorum*. Consequently, we examined the activity of the enzymes encoded by these two genes to identify new steps in the pathway for salvinorin A biosynthesis.

We undertook functional characterization of CYP728D25 and CYP728D27 in *N. benthamiana* by expressing SspKPS and SspKLS from *S. splendens*, and full-length transcripts encoding the cytochrome P450 enzymes SdANS, SdHDAS, and either CYP728D25 or CYP728D27 from *S. divinorum* in pEAQ vectors for transient heterologous expression in leaves of *N. benthamiana*. To improve flux to GGPP, we cloned the AtHMGR gene from *A. thaliana*, and the TccytGGPPS gene from *Taxus canadensis* into pEAQ vectors. Co-infiltration of *Agrobacterium tume-faciens* strains bearing different combinations of these genes resulted in transient expression in *N. benthamiana*. Ethyl acetate extracts of the infiltrated leaves were analyzed by LC-MS in negative mode. In the assays expressing all the known biosynthetic genes plus CYP728D25, two major peaks were detected in extracted ion chromatograms (Fig. 6 and Supplementary Fig. 14), one of *m/z* 331.19 running at 3.1 min and another running at 4.5 min. We tentatively identified the compound eluting at 4.5 min as divinatorin A based on elution time and MS and spectra. In MS$^2$ spectra of divinatorin A, we observed the loss of 44 amu from the molecular ion [M-H]$^-$ *m/z* 331.19 to fragment with *m/z* 287.20 corresponding to loss of carboxylic group $CO_2$, while an MS$^2$ fragment at *m/z* 269.19 corresponded to an additional loss of a water molecule $H_2O$ (18 amu). In addition to these two major MS$^2$ fragments from the peak at 4.5 min, which were present in the same ratio compared to the molecular ion [M-H]$^-$ 331.19, minor MS$^2$ fragments of *m/z* 69.04, 163.08, 191.14, and 245.16 were observed. The identity of this peak was con-firmed by comparison to a divinatorin A standard purified from *Dodonaea viscosa* (Fig. 6 and Supplementary Fig. 14)[58]. This standard

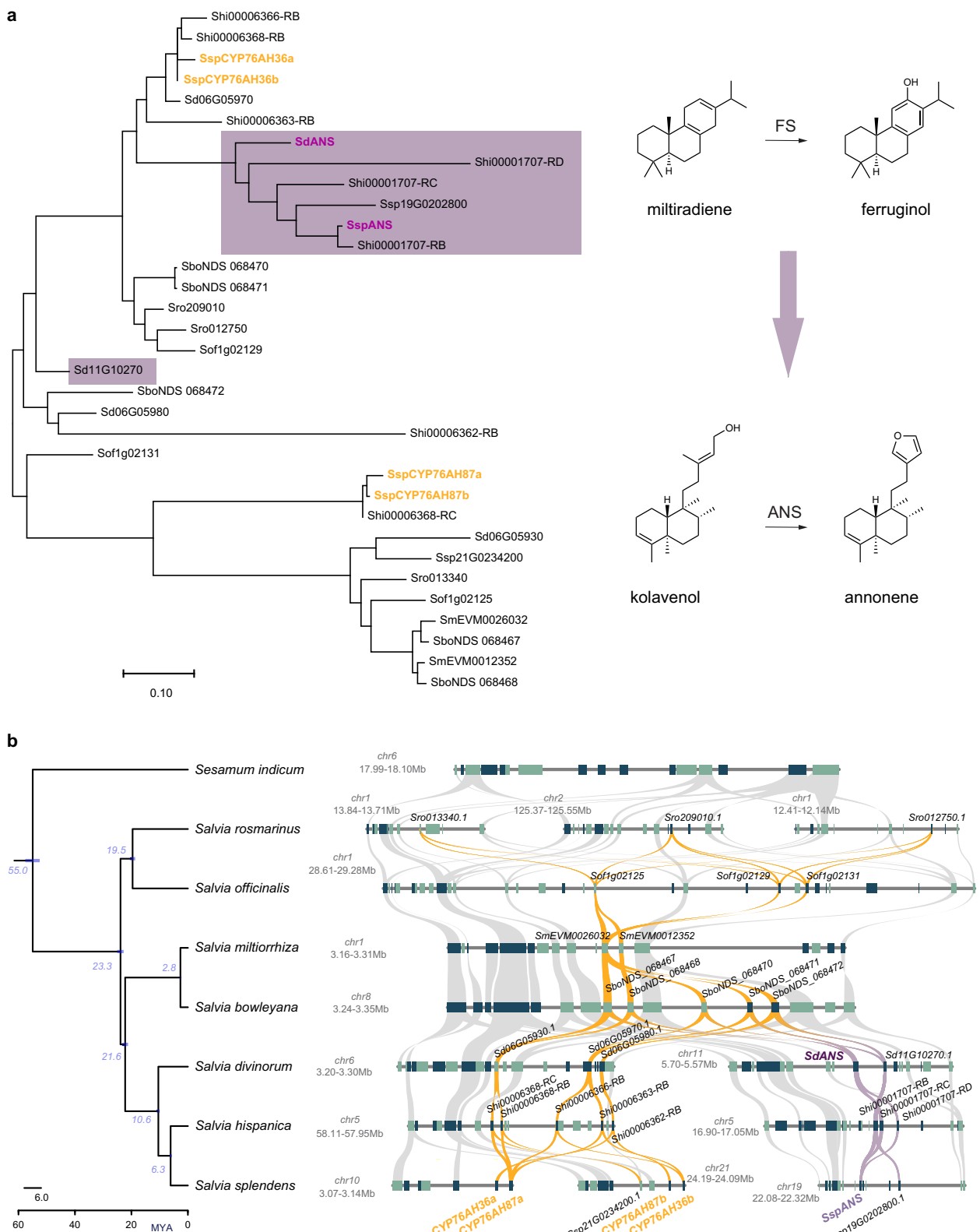

showed one major fragment at $m/z$ 287.2020 in MS$^2$ spectra of molecular ion 331.1919, indicating the loss of a $CO_2$ mass fragment (Fig. 6 and Supplementary Fig. 14), which agreed with previously reported MS$^2$ spectra for divinatorin A[59,60]. Therefore, we decided to refer to CYP728D25 as divinatorin A synthase, SdDAS. However, in the LC-MS analysis of the products of CYP728D25 in *N. benthamiana* leaves, there was an additional peak with $m/z$ 347.19 running at 3.1 min (Fig. 6 and

Supplementary Fig. 15) indicating a second hydroxylation occurred on the hardwickiic acid scaffold, with MS$^2$ fragments at $m/z$ 329.18, 303.20, and 285.19, corresponding to losses from the $m/z$ 331.19 mother ion of a $H_2O$ molecule (18 amu), a carboxylic group $CO_2$ (44 amu), and simultaneous loss of a $H_2O$ molecule and carboxylic group $CO_2$ (18 + 44 amu), respectively. This suggested that CYP728D25/DAS catalyzed a second hydroxylation on the scaffold.

**Fig. 5 | Phylogenetic and phylogenomic analyses of the evolution of annonene synthase activity in *Salvia* species by neofunctionalization of an ancestral gene with ferruginol synthase activity. a** Phylogenetic tree of syntenic genes encoding CYP76AH enzymes in the *Salvia* genus. The purple frame highlights the CYP76AH enzymes that are part of the ferruginol BGC locus in *S. divinorum, S. splendens*, and *S. hispanica*, which arose following WGDα. The characterized CYP76AH enzymes with annonene synthase (ANS) activity are indicated in deep purple bold font, while those with ancestral ferruginol synthase activity in yellow gold bold font. **b** Genomic region and syntenic analysis of the neotropical sages *S. divinorum, S. splendens, S. hispanica*, Asian sages *S. miltiorrhiza*, and *S. bowleyana*, and European sages *S. rosmarinus*, and *S. officinalis*. Gold ribbons highlight the syntenic relationships between CYP76AH genes, while the purple ribbons highlight the duplicated loci which arose after WGDα, containing the neofunctionalized CYP76AH enzymes with annonene synthase activity (SspANS and SdANS) which arose from ancestral genes encoding enzymes with ferruginol synthase activity. Source data are provided as a Source Data file.

Consequently, our working hypothesis for biosynthesis of salvinorins[26] proposes that the pathway proceeds from hardwickiic acid through divinatorin intermediates (Fig. 1a)[26,34,35]. In addition to oxidation on C-18 to form hardwickiic acid and oxidation of C-1 to form divinatorin A, additional oxidations are expected on carbon atoms C-2, C-12, and C-17 of the furanoclerodane scaffold for biosynthesis of salvinorin A (Fig. 1).

We next assayed CYP728D27 in combination with the known biosynthetic genes (but without CYP728D25), in *N. benthamiana*. Two major peaks were detected in extracted ion chromatograms of the LC-MS analysis, one of $m/z$ 331.19 running at 3.2 min and another running at 4.2 min. The MS$^2$ spectra of the peak running at 4.2 min exhibited fragments at $m/z$ 287.20, and 285.19, corresponding to losses from the $m/z$ 331.19 mother ion of a carboxylic group $CO_2$ (44 amu), and simultaneous loss of two hydrogen atoms and carboxylic group $CO_2$ (2 + 44 amu). An additional peak was observed in the extracted ion chromatogram at $m/z$ 347.19 eluting at 3.0 min with MS$^2$ fragments at $m/z$ 303.30 and 301.19, corresponding to losses from $m/z$ 347.19 mother ion of a carboxylic group $CO_2$ (44 amu), and simultaneous loss of two hydrogen atoms and carboxylic group $CO_2$ (2 + 44 amu) (Fig. 6 and Supplementary Fig. 15). Since there were two peaks of $m/z$ 331.19 in the assays of CYP728D27 in *N. benthamiana*, but neither peak ran at the same time as the two peaks for the assays of CYP728D25, nor showed exactly the same MS$^2$ fragments we concluded that CYP728D27 (Fig. 6 and Supplementary Figs. 14 and 15) can catalyze two hydroxylations of the furanoclerodane scaffold but that these are different to the hydroxylations catalyzed by CYP728D25/DAS.

In the LC-MS analysis of assays co-expressing CYP728D25 and CYP728D27 in *N. benthamiana*, three major peaks were observed in the extracted ion chromatogram at $m/z$ 347.19. The peaks eluting at 3.0 and 3.1 min have been observed before in assays of CYP728D27 and CYP728D25 (Fig. 6 and Supplementary Fig. 15). A new peak eluting at 3.9 min was observed (Fig. 6f and Supplementary Fig. 15) with MS$^2$ fragments of $m/z$ 329.18, 303.20, and 285.19, corresponding to losses from $m/z$ 347.19 mother ion of a $H_2O$ molecule (18 amu), carboxylic group $CO_2$ (44 amu), and simultaneous loss of a $H_2O$ molecule and carboxylic group $CO_2$ (18 + 44 amu). Salvinorin A precursors of divinatorin B with hydroxyl groups on C-1 and C-17, and divinatorin F with hydroxyl groups on C-1, C-2, and C-17 may be formed by the combined activities of CYP728D25, CYP728D26, and CYP728D27. Since the hydroxylation at C-17 was most prevalent we can assume that one of the observed peaks with $m/z$ 331.19 corresponds to mono-hydroxylated hardwickiic acid (hydroxylated at C-17 - proposed intermediate 2), and one of the of the observed peaks with $m/z$ 347.19 corresponds to the di-hydroxylated product of hardwickiic acid (hydroxylated at C-17 in addition to C-1 - proposed intermediate 1) (Fig. 1).

**Tracing the occurrence of the genes encoding CYP728D hydroxylases in *S. divinorum* within the genus *Salvia***

Following a similar analysis to that we had used for the CYP76AH subfamily genes in different *Salvia* species to examine the origins of annonene synthase, we aligned the syntenic analysis of CYP728D subfamily genes according to the species alignments on a phylogenetic tree (Fig. 7a, extended synteny and Supplementary Fig. 16). While in the syntenic genomic area of *S. rosmarinus* there are no genes encoding CYP728D proteins, there is one such gene in the syntenic genomic areas of *S. officinalis* and *S. bolweyana*. There are three such genes in the syntenic genomic area of *S. miltiorrhiza*, which cluster closely together on the phylogenetic tree, indicating recent tandem duplications following the divergence of *S. miltiorrhiza* from *S. bolweyana* (2.8 MYA). In *S. hispanica* there are four gene copies in the syntenic genomic area resulting from tandem gene duplications, while in *S. splendens* there are two syntenic genomic areas likely resulting from the recent WGDβ with the previously identified genes encoding CYP728D62 (SsHDAS), CYP728D63 (SsHTAS), and CYP728D64[28] located in the same genomic region. In the syntenic genomic region of *S. divinorum* there are four genes, including CYP728D25 (SdDAS), CYP728D26 (SdHDAS), and CYP728D27, located on chromosome 3. In both *S. divinorum* and *S. splendens*, orthologous CYP728D enzymes catalyze the oxidation of the methyl C-18 of the clerodane scaffold to carboxylic acid, and paralogous CYP728D enzymes act on C-19 in *S. splendens* and on C-1, C-2, C-12, and C-17 in *S. divinorum* (Fig. 7b). Phylogenetic analysis supported further these findings and showed that although the CYP728D proteins in *S. divinorum* are part of the same clade as those active in other Calosphace species, the three enzymes cluster closely and they do not have identical activities to the CYP728D enzymes from other species such as the CYP728Ds in *S. splendens* and *S. hispanica*. We hypothesize that HDAS activity evolved first, and after the divergence of *S. divinorum* from the rest of the subgenus, lineage-specific gene duplications took place, facilitating subfunctionalisations that resulted in the evolution of the pathway towards salvinorin A biosynthesis. Our phylogenetic analysis of CYP728D proteins indicated that the neofunctionalization of HDAS to HTAS in *S. splendens* was the result of a tandem gene duplication, shared by *S. splendens and S. hispanica* and supported by the presence of clerodane compounds in *S. hispanica* with hydroxyl groups on C-19 of the clerodane scaffold, such as salvihispinin A, and hispanin F[61,62].

**A carboxy methyl transferase of the SABATH-family acts to complete divinatorin B biosynthesis**

The catalytic activities of the family CYP728D enzymes from *S. divinorum* result in the synthesis of precursors of divinatorin B and F (Fig. 6), with a methylation of the carboxyl group required for the synthesis of divinatorin B from divinatorin A.

A methyltransferase, highly expressed in *S. divinorum* trichomes, was reported in the PhD thesis of L. Kutrzeba (University of Mississippi)[34]. This enzyme showed activity in vitro in biochemical assays using radiolabelled SAM, divinatorin A, and hardwickiic acid as substrates[34]. Consequently, we searched the transcriptomic data from trichomes to identify the most highly expressed transcripts in trichomes, annotated functionally as methyl transferases. Three genes encoding SAMT- dependent methyl transferases were highly expressed: *Sdiv09G23780*, an annotated carboxy methyl transferase, and two tandemly duplicated genes *Sdiv03G06170* and *Sdiv03G06180*. Assays in vivo in yeast of these genes with co-expression of the upstream genes *SspKPS*, *SspKLS*, *SdANS*, *SdHDAS*, and *SdDAS* failed to detect any methylation activity. Consequently, we switched to transient expression in *N. benthamiana* and co-infiltrated *Agrobacterium* strains bearing the three candidate methyl transferases, with those expressing AtHMGR, TcGGPPS, SspKPS, SspKLS, SdANS, SdHDAS, and SdDAS/CYP728D25 (Fig. 8b).

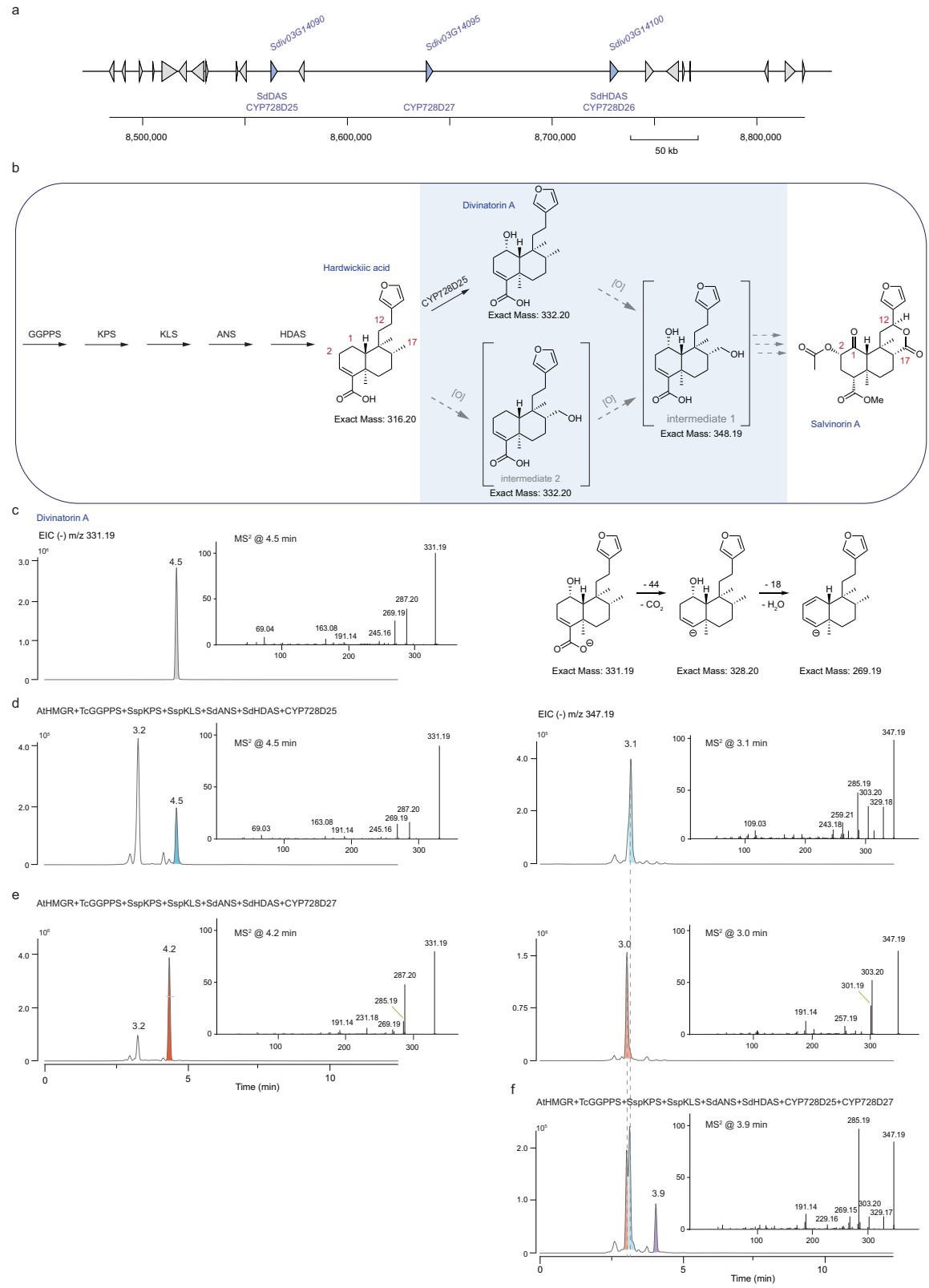

Analysis of ethyl acetate extracts of infiltrated *N. benthamiana* leaves was performed using LC-MS in positive mode. Only the product of gene *Sdiv09G23780* showed enzymatic activity towards the hydroxylated clerodanes (Fig. 8b), with one major peak which appeared consistently in all biological replicates, running at 4.2 min with a molecular ion $[M + H]^+$ at *m/z* 363.22. The molecular mass of this peak corresponded to the predicted masses of molecular ions of divinatorin

B. The gene encoding the methyl transferase[34] was identical to the *Sdiv09G23780* gene. This indicated that the protein encoded by *Sdiv09G23780* has carboxy methyl transferase activity, and we called it SdCMT.

Likewise, we assayed the CYP728D27 in combination with CYP728D25 (SsDAS) and SdCMT, however no new peak appeared while the peak at 4.2 min with a molecular ion $[M + H]^+$ at *m/z* 363.22

**Fig. 6 | Combined expression of S. divinorum cytochrome P450 enzymes CYP728D25 and CYP728D27with upstream enzymes in furanoclerodane metabolism. a** Genomic region (chromosome 3) of the *S. divinorum* genome showing that genes Sdiv03G14090, Sdiv03G14095, and Sdiv03G14100 encoding CYP728D25, CYP728D27, and CYP728D26, respectively, are the results of tandem duplications. **b** The furanoclerodane biosynthetic pathway from the synthesis of geranylgeranyl diphosphate to hardwickiic acid is shown in the white frame. The blue frame depicts the oxidations observed on the hardwickiic acid scaffold after co-expression of CYP728D25 with the upstream genes in *N. benthamiana*. **c** LC-MS analysis of divinatorin A standard isolated from *Dodonaea viscosa* (selected *m/z* signal of 331.19 in negative mode). MS$^2$ spectra of the peak at 4.5 min. gave fragments of *m/z* 331.19 ion with 45% high-energy collisional dissociation (HCD) and fragmentation of divinatorin A, losing a CO2 fragment (44 amu), followed by the loss of a water molecule (18 amu). **d** LC-MS analysis of extracts (selected *m/z* signals of 331.19 and 347.19 in negative mode) of infiltrated *N. benthamiana* leaves with

Agrobacterium strains carrying AtHMGR, TcGGPPS, SspKPS, SspKLS, SdANS, and SdHDAS with CYP728D25 (SdDAS). MS$^2$ spectra of peaks eluting at 3.1 (from 347.19 mother ion) and 4.5 min (from 331.19 mother ion) are shown. **e** LC-MS analysis of extracts (selected *m/z* signals of 331.19 and 347.19 in negative mode) of infiltrated *N. benthamiana* leaves with Agrobacterium strains carrying AtHMGR, TcGGPPS, SspKPS, SspKLS, SdANS, and SdHDAS with CYP728D27. MS$^2$ spectra of peaks eluting at 4.2 (from 331.19 mother ion) and 3.0 min (from 347.19 mother ion) are shown. **f** LC-MS analysis of extracts (selected *m/z* signals of 347.19 in negative mode) of infiltrated *N. benthamiana* leaves with Agrobacterium strains carrying AtHMGR, TcGGPPS, SspKPS, SspKLS, SdANS, and SdHDAS with CYP728D25 (SdDAS) and CYP728D27. The two peaks running at 3.1 min for CYP728D25 and 3.0 min for CYP728D27 are apparent, but an extra peak running at 3.9 min was observed. MS$^2$ spectra of the peak eluting at 3.9 min (from 347.19 mother ion) are shown.

disappeared (purple trace Fig. 8b). This significant reduction in the methylated product of SdCMT suggested that hydroxylations catalyzed by CYP728D27 inhibit the formation of carboxy-methylated compounds and thus divinatorin B, probably due to substrate specificity of the CMT. These data also indicated that the second hydroxylation of divinatorin A by CYP728D25 was on C-17 to produce intermediate 1, and consequently, the product of CYP728D25 and SdCMT is divinatorin B. In addition, although only proof by elimination, the hydroxylations catalyzed by CYP728D27 are likely to be on C-2 and C-12, while the substrate specificity of the CMT would suggest that CYP728D27 works on divinatorin B to produce divinatorins E and F.

*SdCMT* is located on chromosome 9 of the *S. divinorum* genome (Fig. 9a) together with four additional genes encoding methyl transferases. Syntenic analysis (Fig. 9a) showed that the presence of this gene in this specific genomic area has been conserved at least since the MRCA of the genus *Salvia* and *Sesamum indicum*. While in *S. officinalis, S. miltiorrhiza*, and *S. bowleyana* there is just one genomic area in the synteny, there are two syntenic genomic regions in *S. divinorum* on chromosomes 9 and 2, two syntenic regions in *S. hispanica* on chromosomes 1 and 5, and three syntenic regions in *S. splendens* on chromosomes 13, 14 and 22. These multiple syntenic regions in the genomes of the three species in the subgenus Calosphace, reflect the effects of WGDα and WGDβ on these genomes. However, only in *S. divinorum* in the syntenic region on chromosome 9 is there genetic enrichment of *CMT* genes, suggesting that there were tandem gene duplications following the divergence of *S. divinorum*, 10.6 MYA. The methylation of the C-18 carboxyl functional group has been reported only in clerodanes from *S. divinorum* and not in clerodanes from *S. splendens* or *S. hispanica*[61–63], suggesting that the duplication of *CMT* genes on chromosome 9 may have been key to the evolution of savinatorin A biosynthesis in *S. divinorum*.

SABATH methyltransferases are a diverse family of SAMT-dependent methyl transferase enzymes in plants that catalyze the methylation of carboxyl functional groups using SAMT as a cofactor and a methyl donor[64]. Members of the family play pivotal roles in modulating plant hormones and in several branches of specialized metabolism. We used examples of SABATH-family enzymes from plants (Supplementary Table 8) to construct a phylogenetic tree that included the SdCMT (Fig. 9b, extended Phylogenetic tree and Supplementary Fig. 17) and its syntenic pairs. The closest enzyme, phylogenetically, to SdCMT was the farnesoic acid methyl transferase AtFAMT from *A. thaliana*[65] suggesting that this type of methyl transferase might have particular affinity for substrates of terpene origin.

## Discussion

*S. divinorum* produces the non-nitrogenous κ-opioid receptor agonist salvinorin A, a furanoclerodane diterpenoid with strong hallucinogenic effects and considerable pharmacological interest. By combining a chromosome-level genome assembly with comparative genomics and targeted biochemical assays, our work clarifies how salvinorin A

biosynthesis emerged within the genus *Salvia* and identifies additional enzymes that extend current understanding of furanoclerodane and diterpenoid metabolism. The high-quality *S. divinorum* genome (541.9 Mb; BUSCO completeness 98.8%) provides a robust framework for tracing the microevolution of specialized metabolism of clerodanes in neotropical sages. Comparative analyses across American, Asian, and European *Salvia* species confirm that pathway innovation has been strongly shaped by whole-genome duplication events followed by lineage-specific tandem duplications. This pattern mirrors broader trends in specialized metabolism in the Lamiaceae, where gene duplication supplies raw material for enzymatic diversification[36]. While our analyses reveal gene family expansions and WGD events in *Salvia* subgenera (Fig. 3), these insights are derived from publicly available genomes, and potential differences in assembly quality should be considered when interpreting syntenic relationships.

Early steps of clerodane biosynthesis seem to be conserved at the genome level[25]. *SdKPS* resides in a syntenic region in common with class II diterpene synthases in other species, which were derived from an ancient WGD duplication of a genomic locus with an *ent*-copalyl diphosphate synthase, reinforcing earlier biochemical evidence that subtle active-site mutations of a key conserved histidine residue to an aromatic amino acid−tyrosine or phenylalanine−underlie the emergence of kolavenyl diphosphate synthase activity[26,29,66]. Among class I diterpene synthases, *SdKSL2* shows expression patterns and syntenic relationships indicating a gene functionality arrangement specific to the genus *Salvia*, highlighting functional partitioning following duplication.

Genes encoding cytochrome P450 enzymes have been expanded and diversified functionally in *S. divinorum*. Comparative phylogenomics indicated that annonene synthase activity arose after a whole-genome duplication shared by the American *Salvia* species, involving neofunctionalization of ancestral ferruginol synthases. Subsequent lineage-specific duplications within the CYP728D subfamily enabled further oxidation of the clerodane scaffold. Functional assays demonstrated that CYP728D25 acts as a divinatorin A synthase (SdDAS), and can also add a second hydroxyl group to the furanoclerodane skeleton, probably at C-17 to facilitate the synthesis of divinatorin B in combination with SdCMT. The related paralog CYP728D27 contributes additional oxidations at multiple positions (likely at C-2 and C-12). Collectively, the activity of the three enzymes (CYP728D25/SdDAS, SdCMT and CYP726D27) reported here could account for the synthesis of all the intermediates identified as divinatorins in *S. divinorum* (except for divinatorin D, which would require the additional activity of an acetyl transferase) along the pathway to salvinorin A biosynthesis (Fig. 1). This illustrates how subfunctionalization and catalytic promiscuity working with a network of intermediates can drive pathway elaboration.

Methylation of the C-18 carboxyl group represents a key step in divinatorin biosynthesis, and a modification unique to clerodanes

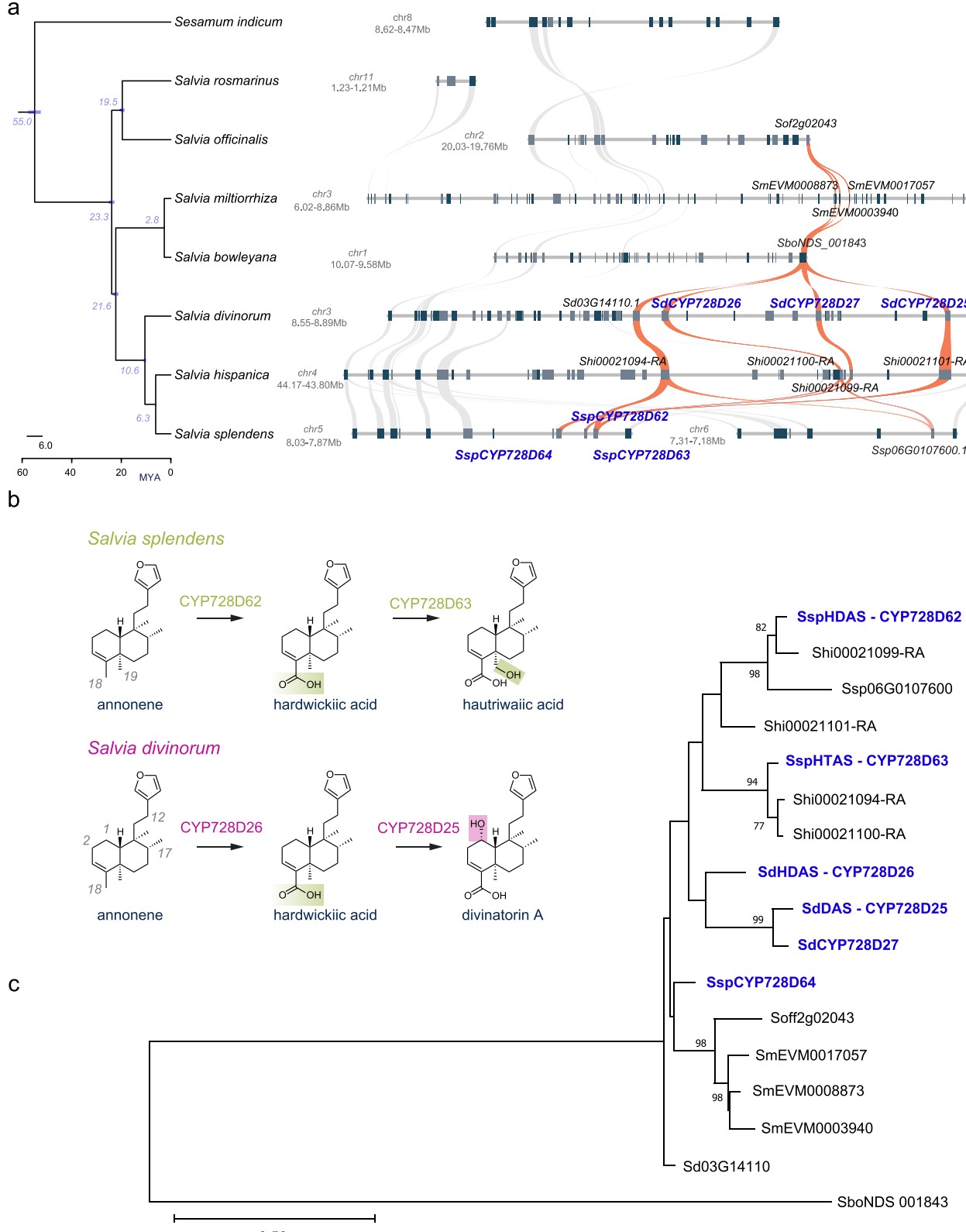

**Fig. 7 | Phylogenetic and phylogenomic analyses of the evolution of CYP728D enzymatic activity in *Salvia* species. a** Genomic region and syntenic analysis of the genes encoding CYP728D enzymes in the neotropical sages *S. divinorum*, *S. splendens*, *S. hispanica*, Asian sages *S. miltiorrhiza*, and *S. bowleyana*, and European sages *S. rosmarinus*, and *S. officinalis*. Orange ribbons highlight the syntenic relationships between CYP728D genes. **b** Different catalytic functionalities and oxidation sites on the furanoclerdane scaffold catalyzed by the CYP728D enzymes in *S.* *divinorum* and *S. splendens*. In green are highlighted the oxidation sites of *S. splendens* CYP728Ds, and in purple are highlighted the oxidation sites of *S. divinorum* CYP728Ds. In mixed color is highlighted the common catalytic functionality for the oxidation of C-18. **c** Phylogenetic tree of syntenic CYP728D proteins in the *Salvia* genus. Blue bold font has been used to highlight the CYP728D enzymes assayed from *S. divinorum* and *S. splendens*. Source data are provided as a Source Data file.

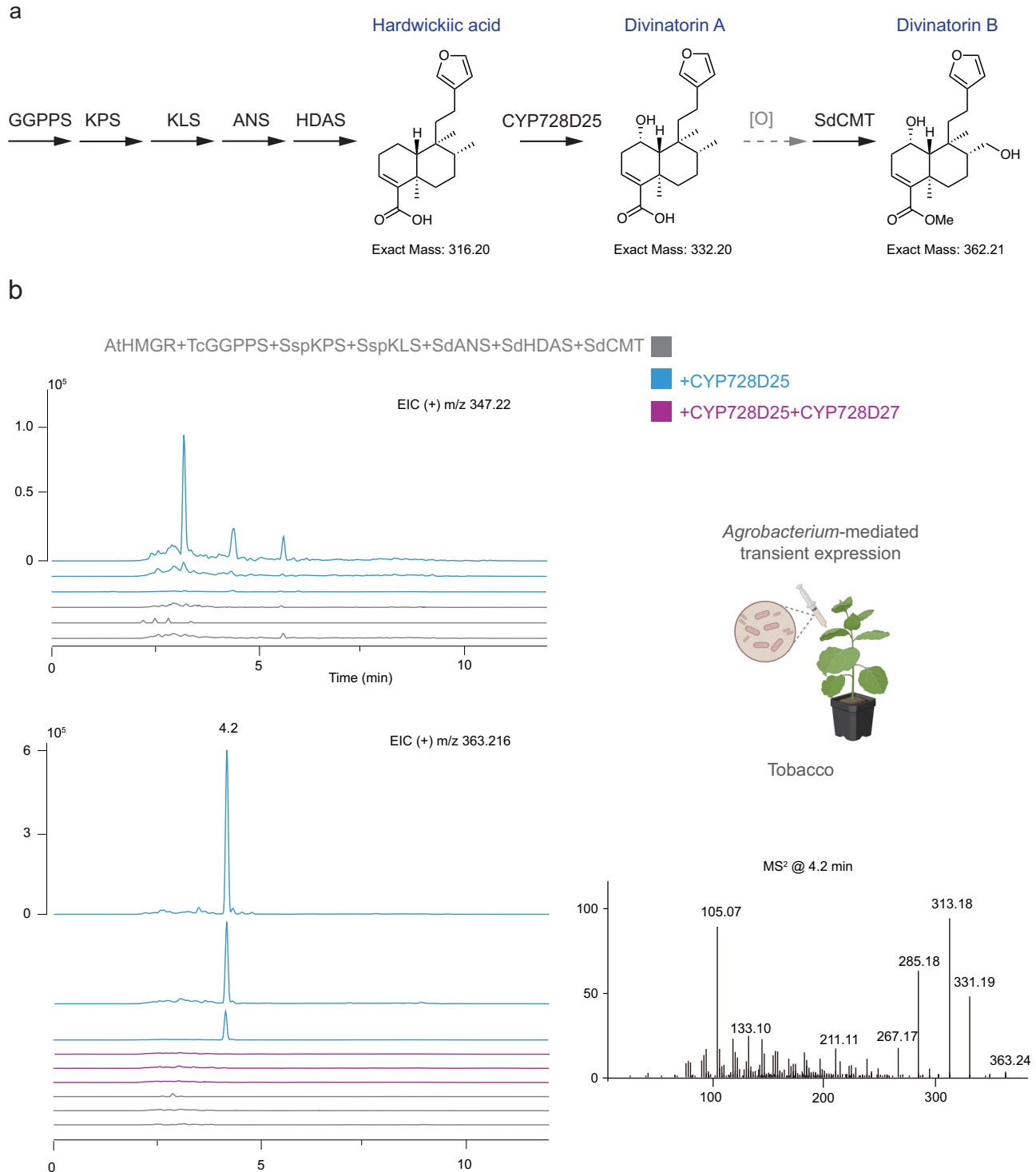

**Fig. 8 | Transient expression of *S. divinorum* cytochrome SABATH-family methyl transferase SdCMT with upstream enzymes of furanoclerodane metabolism in *N. benthamiana*. a** The furanoclerodane biosynthetic pathway from the synthesis of geranylgeranyl diphosphate to divinatorin B. The gray dashed arrows depict the observed oxidations on the hardwickiic acid scaffold following the co-expression of CYP728D25 and SdCMT in *N. benthamiana* together with the upstream genes. **b** LC-MS analysis of extracts (selected *m/z* signals of 347.22 and 363.216, in positive mode) of infiltrated *N. benthamiana* leaves with *Agrobacterium* strains carrying AtHMGR, TcGGPPS, SspKPS, SspKLS, SdANS, SdHDAS, and SdCMT with CYP728D25 (SdDAS) (blue traces) or with CYP728D25 (SdDAS) and CYP728D27 (without (gray traces). MS² spectra of peak eluting at 4.2 min. The three traces of each combined infiltration correspond to three separate biological repetitions.

from *S. divinorum*. Identification and functional validation of the activity of the SABATH-family carboxy methyltransferase, SdCMT, reveal that this step likely evolved through tandem gene duplication after divergence from closely related species. The restriction of this activity to *S. divinorum* illustrates how confined gene expansions can underpin species-specific chemical traits.

Together, these results show that salvinorin A biosynthesis arose through sequential recruitment and diversification of enzymes from pre-existing terpenoid pathways. The integration of genome-scale

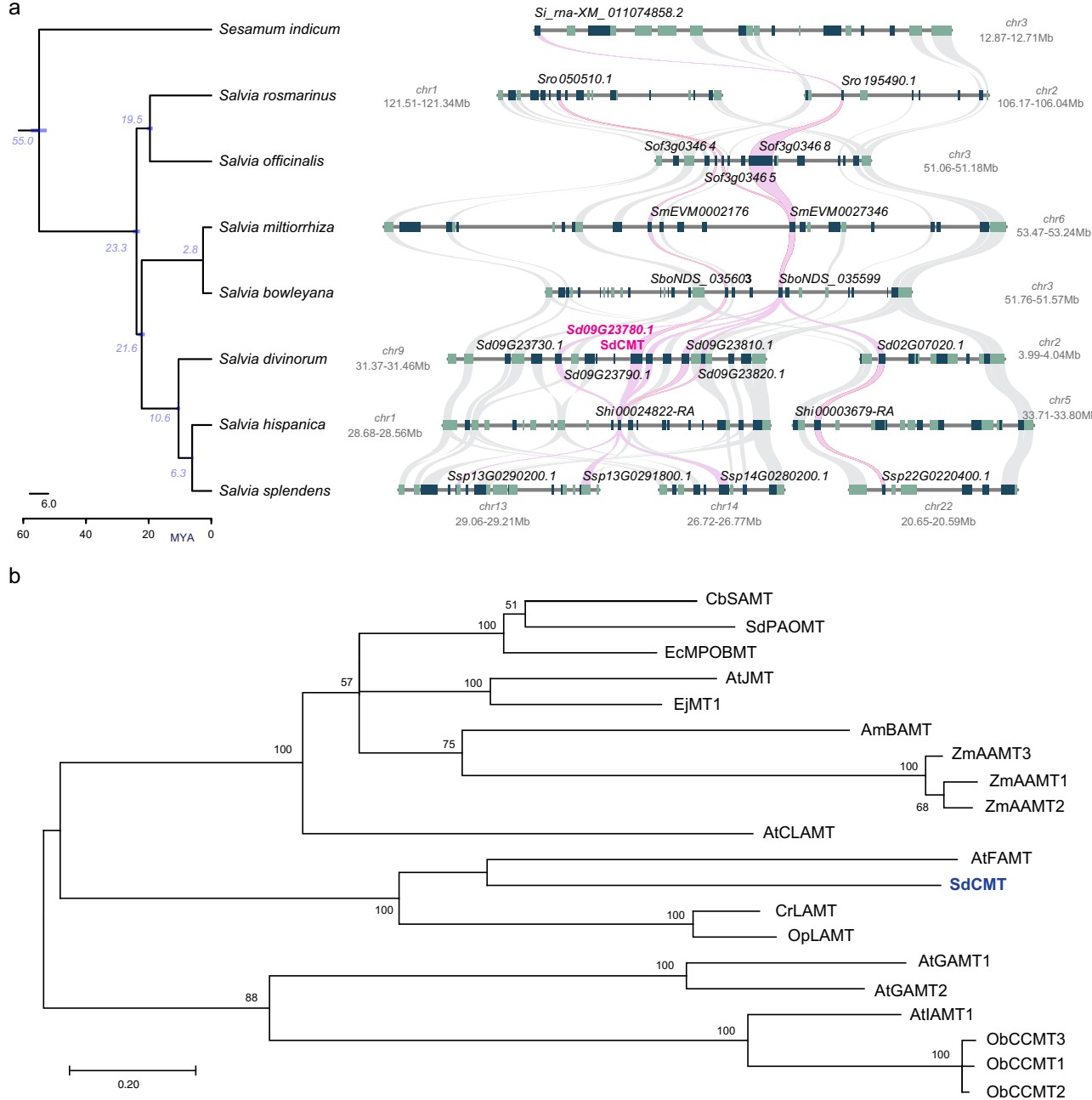

**Fig. 9 | Phylogenetic and phylogenomic analyses of the SdCMT in *Salvia* species. a** Genomic region and syntenic analysis of SdCMT in the neotropical sages S. *divinorum*, *S. splendens*, *S. hispanica*, Asian sages *S. miltiorrhiza*, and *S. bowleyana*, and European sages *S. rosmarinus*, and *S. officinalis*. Purple ribbons highlight the syntenic relationships between SdCMT homologous genes. **b** Phylogenetic tree of functionally characterized SABATH-family plant methyl transferases (mainly carboxy methyl transferases). In blue bold font is highlighted the gene encoding SdCMT in *S. divinorum*. Source data are provided as a Source Data file.

analyses with heterologous expression systems highlights *N. benthamiana* as a particularly effective system for resolving late-stage pathway steps that are difficult to reconstruct in yeast. While several oxidations and acylation reactions remain to be assigned, the genomic and evolutionary signatures identified here provide clear criteria—trichome-enriched expression, gene family expansion, and tandem duplication events—for prioritizing additional candidate enzymes. The recent example of complete hyperforin biosynthetic pathway elucidation by single-cell RNA-seq[67] provides an additional tool to further the study of confined metabolic pathways in specialized multicellular structures, such as glandular trichomes. Overall, this work establishes a genomic and biochemical framework for completing the salvinorin A pathway and for engineering furanoclerodane biosynthesis in the future. Beyond salvinorin A and *S. divinorum*, this work illustrates how comparative genomics at the microevolutionary scale can reveal the mechanisms by which plants generate chemically and pharmacologically distinctive specialized metabolites.

## Methods

### Plants

*S. divinorum* plants were purchased from Enchanted Plants Nursery (https://enchantedplantsnursery.co.uk/) and were cultivated in accordance with all international conventions, including the Nagoya Protocol (https://www.cbd.int/abs/default.shtml).

## Flow cytometry analysis

About 0.2 g fresh *S. divinorum* leaves were cut and collected, chopped into small pieces with a sharp blade and saturated in 500 µL Nuclei Extraction buffer. After 60 s, the liquid buffer was filtered through a 50-µm strainer and mixed with 2 mL staining buffer (Sysmex CyStainPI Absolute P kit) containing PI (propidium iodide) and RNase for 30 minutes in the dark. The suspension was analyzed by CyFlow Cube6 Flow Cytometer (Sysmex Partec, Muenster, Germany) and the corresponding FCSExpress software. *Solanum lycopersicum* was used as an internal standard. According to the C-value database, the genome size of the *Salvia* genus ranges from 440 Mb to 2 Gb, with chromosome numbers ranging from 7 to 11 and ploidy levels from 2 to 8. The 1 C value of *Solanum lycopersicum* (tomato) is 1.06 pg and, based on the following formula:

$$
\begin{aligned}
&\text{Sample genome value}/\text{genome value} \\
&= \text{Sample peak fluorescence intensity}/ \quad\quad (1) \\
&\text{Reference fluorescence intensity}
\end{aligned}
$$

We calculated that the 1 C value of *S. divinorum* is 0.548 pg (Supplementary Table 1). Given that 1 pg is equivalent to 980 Mb, based on the 1 C value of *S. divinorum*, the genome size was calculated as follows $0.548 \times 980 \, \text{Mb} \approx 537.44 \, \text{Mb}$.

## DNA nano balls (DNB) short-read sequencing

Genomic DNA from *S. divinorum* leaves was extracted with 2% CTAB lysis solution, supplemented with 2% beta-mercaptoethanol. The library preparation was carried out using the VAHTSTM Universal DNA Library Prep Kit for MGI (VAZYME, China). Genomic DNA was fragmented and then converted to blunt-end DNA through an end repair reaction. An adenosine was added to the 3′ end via an A-tailing reaction. Next, library adapters were ligated to both ends of the DNA. The resulting library products underwent amplification via PCR and were assessed for quality.

Subsequently, the double-stranded library products were denatured to produce single-stranded DNA, which was then circularized. Linear single-stranded DNA was removed by digestion. The final single-stranded circularized library was amplified using phi29 and rolling circle amplification (RCA) to create DNA nano balls (DNBs), each containing approximately 300 copies of the initial single-stranded library molecule. These DNBs were placed into a patterned nanoarray, where sequencing reads of PE100/150 bases were generated using the DNBSEQ-G400/T7/T10 platform (BGI-Shenzhen, China).

## PacBio sequencing

For long-read sequencing, extracted genomic DNA was sheared using a Megaruptor to ~15–20 kb. The DNA then underwent damage repair, end repair, ligation with specific adapters, enzymatic digestion, and size selection, resulting in a SMRTbell library. The qualified library was sequenced on the PacBio platform. HiFi reads with long-read lengths (10-20 kb) and high accuracy (greater than 99% accuracy) were produced using the Circular Consensus Sequencing (CCS) method.

## HiC library sequencing

Genomic DNA from *S. divinorum* leaves was crosslinked using 4% formaldehyde, enabling the binding of DNA and adjacent proteins. This cross-linked DNA was then digested with the restriction enzyme *Mbo* I, releasing the ends of the linear fragments, which were tagged with biotin and ligated. The ligated DNA from the crosslinked complexes was extracted, and biotin-labeled DNA regions were captured. Finally, general library construction and sequencing were performed to obtain the locus information from both ends of the captured sequences. After adding an "A" base to the 3′ end of each strand, single-stranded PCR products were produced via denaturation, followed by the configuration of the circularization system. Rolling circle amplification

produced multiple copies of single-stranded circular DNA, forming DNA nano balls (DNBs). High-quality DNBs were then loaded into patterned nanoarrays using a high-intensity DNA nanochip technique and sequenced through combinatorial Probe-Anchor Synthesis (cPAS).

## Genome survey

Survey analyses of the *S. divinorum* genome were conducted using GCE (v1.0.2)[68], Jellyfish (v2.3.0)[69], GenomeScope[8,70], GenomeScope2 (v2.0.1)[40], and Smudgeplot (v0.2.3)[40] based on Illumina sequencing data. For GCE analysis, genome characteristics were estimated by modeling the k-mer frequency distribution with a Poisson model. K-mer sizes of 17 and 19 were selected, and their total counts and coverage were calculated using kmerfreq, followed by genome size, heterozygosity, and repeat content estimation with GCE.

Jellyfish was used to generate k-mer frequency distributions with k-mer sizes of 17, 19, 21, and 23. Genome size, repeat content, and heterozygosity were estimated with GenomeScope, and further evaluated with GenomeScope2 using the same histograms under multiple ploidy assumptions (2, 3, and 4) for each k-mer size.

Ploidy levels and genomic complexity were inferred with Smudgeplot based on 17-mer histograms from Jellyfish. The lower (L) and upper (U) k-mer coverage cut-off values were determined from the histogram using the "smudgeplot.py cutoff" command, following the developer's recommendations. K-mers within this coverage range were extracted with "jellyfish dump" and used to generate heterozygous k-mer pair counts (smudgeplot.py hetkmers), which were then visualized with "smudgeplot.py plot".

## Genome assembly and genome comparisons

The third-generation sequencing data of *S. divinorum* were assembled using Hifiasm v0.19.9[41], resulting in a preliminary assembly size of 574,586,498 bp (410 contigs). The assembly was then deduplicated using Purge_Dups[42], yielding a final assembly size of 558,956,549 bp (215 contigs). HiC clean data were aligned to the contig sequences, and the contigs were scaffolded using juicer-3Ddna (20190319)[43]. A total of 20 super scaffolds (541,872,582 bp) were anchored to 11 chromosomes.

The previously published *S. divinorum* genome sequence[43] was downloaded from NCBI (PRJNA1104206), and this version is discontinuous. The NCBI genome version (NCBI.fasta) and our assembled genome version (Salvia_divinorum.chr.fa) were compared using the following software and command lines. Nucmer, delta-filter, show-coords and mummerplot in the Mummer[71] toolkit were used for the genome comparison. Structural identification of the alignment results was performed by using syri[71,72] with the command line: syri -c ref_qry.delta.filter.coords -d ref_qry.delta.filter -r NCBI.fasta -q Salvia_divinorum.chr.fa. Structural variation plots were constructed using plotsr[73] with the following command line: plotsr --sr syri.out --genomes genomes.txt -H 8 -W 5 -b pdf.

## Annotation of repetitive DNA sequences

Repeat sequence annotation was divided into two main parts: known and de novo prediction. Known prediction involved using a known repeat sequence database to identify repeat regions in the genome. The database used was RepBase (v23.06, https://www.girinst.org/repbase/)[74], and the prediction software was RepeatMasker (v4.1.6)[75]. De novo prediction involved building a de novo repeat library for the genome using ltr_finder (v1.0.6)[76], and RepeatModeler 2.0.5[77]. After filtering and classification, the final de novo database was obtained, and RepeatMasker (v4.1.6)[78] was used to annotate repeat regions in the genome.

## Gene prediction and annotation

Gene prediction was performed using MAKER (v3.01.03)[79]. In the first round, transcriptome and protein data were used for gene prediction,

with MAKER parameters set to est2genome = 1 and protein2genome = 1. From the prediction results, 2000 structurally complete genes (with start and stop codons, and no premature termination or frameshifts) were randomly selected. In the second round. Augustus and SNAP were used to train parameters using these 2000 structurally complete genes. The transcriptome prediction results, protein results, and two de novo prediction results were integrated to obtain higher-quality gene model predictions. The datasets used for MAKER annotation in this project included transcriptome data and homologous species protein data (*A. thaliana*, *S. divinorum*, *S. miltiorrhiza*, *Sesamum indicum*, *S. hispanica*, *S. splendens*, *Solanum lycopersicum*). The de novo training software used was Augustus (v3.5.0)[80], and SNAP (20060728)[81]. Transcriptome data were first aligned using hisat2 (v2.2.1)[82], and then transcripts were constructed using StringTie (v2.2.1)[83]. Finally, MAKER was used to integrate the results and obtain the final gene set.

The annotation of gene function involved comparing the gene set obtained from the gene structure annotation with known protein databases and other libraries using the alignment software Blast (v2.2.31)[84], to obtain functional information. The databases used for this method included SwissProt, TrEMBL, and NR. InterProscan (v5.71-102.0)[85], and were used to search secondary structure domain databases for information on gene function, using databases such as SUPERFAMILY, NCBIFAM, PRINTS, Pfam, SMART, ProSiteProfiles, and ProSitePatterns. KEGG pathway annotation was performed using eggnog-mapper (v2.1.12)[86], with the database emapperdb (v5.0.2).

### Annotation of non-coding RNA
Non-coding RNAs (ncRNAs), which do not translate into proteins, include rRNA, tRNA, snRNA, miRNA, and these RNAs have important biological functions. tRNAscan-SE (v2.0, http://lowelab.ucsc.edu/tRNAscan-SE/)[87] was used to identify tRNA sequences in the genome based on their structural features. Due to the high conservation of rRNA, rRNA sequences from closely related species were used as reference sequences, and BLASTN[84] was used to identify rRNA sequences in the genome. The covariance models from the Rfam family (v14.10)[88], and the INFERNAL (v1.1.5, http://infernal.janelia.org/)[89] included in Rfam were used to predict miRNA and snRNA sequences in the genome.

### Species divergence time estimation
Ortholog clustering was performed using OrthoFinder v2.5.4 on the genomes of *S. divinorum*, *S. rosmarinus* (https://doi.org/10.6084/m9.figshare.21443223.v1), *S. bowleyana* (National Genomics Data Center: PRJCA003734), *S. miltiorrhiza* (National Genomics Data Center: PRJCA003150), *S. hispanica* (China National GeneBank Database: CNA0047366), *S. splendens* (PRJNA422035), *S. officinalis* (PRJCA010537, accession number: GWHBJVP00000000), *Sesamum indicum* (PRJNA186669), *Solanum lycopersicum* (SL3.0, EnsemblPlants), *A. thaliana* (TAIR10, EnsemblPlants), and *Andrographis paniculata* (PRJNA421867). Protein sequences were filtered to keep only the longest isoforms, discarding those shorter than 60 amino acids. The refined sets were analyzed by OrthoFinder with settings for multiple sequence alignment using MAFFT, sequence search via DIAMOND, and tree inference with FastTree. Speciation times were estimated with mcmctree using parameters: burnin 5000000, sampfreq 50, and nsample 1000000 from PAML, based on the phylogenetic tree generated by OrthoFinder and calibrated with divergence times from TimeTree[90], including divergence estimates for *S. miltiorrhiza* and *S. officinalis* (18–28.9 MYA), *S. lycopersicum* and *A. paniculata* (75.8–96.6 MYA), and *S. lycopersicum* and *A. thaliana* (111.4–123.9 MYA).

### Gene family evolution
Gene family size changes were assessed using CAFE v5.1.0 (https://github.com/hahnlab/CAFE) to determine the expansions and contractions of gene families in each species. The analysis of shared and unique gene families between *S. divinorum*, *S. rosmarinus*, *S. bowleyana*, *S. miltiorrhiza*, *S. hispanica*, *S. splendens*, *S. officinalis*, *Sesamum indicum*, *Solanum lycopersicum*, *A. thaliana*, and *Andrographis paniculata* was performed and visualized using the modified UpSetR package (https://github.com/GuangchuangYu/UpSetR).

### Synteny analysis
The species sequences used in phylogenetic analysis were imported into MCscan (Python version), a tool from JCVI utility libraries (https://github.com/tanghaibao/jcvi). Gene pair comparisons were conducted using LAST. After removing potential tandem duplications and low-scoring hits, the LAST output anchors were grouped into syntenic blocks. Macrosynteny plots, microsynteny plots and dot plots were generated using dotplot, karyotype, and synteny functions with default settings.

### Phylogeny of CYP450 and CMT genes
CYP450 protein sequences were chosen based on syntenic analysis, while CMTs were selected from the literature. Protein sequences were aligned using ClustalW in MEGA12[91]. The best-fit substitution models were JTT + I + G for CYP76AH, JTT + I + G + F for CYP728D, and LG + G + I for CMTs, each with a bootstrap value of 1000. Finally, the phylogenetic trees were constructed using the maximum likelihood method.

### Speciation and WGD analysis
Homologous gene pairs were identified through BLASTP analysis of the proteomes from *S. splendens*, *S. hispanica*, and *S. divinorum*. MCScanX was used to infer syntenic genes both within and across genomes, utilizing gene similarity and gene order data. The synonymous substitution rate ($Ks$) for the gene pairs was calculated using KaKs_calculator 2.0[92]. To estimate the time of WGD events we used the following formula

$$T = Ks/2\lambda \qquad (2)$$

where $\lambda$ is $6.5 \times 10^{-9}$, the average substitution rate for dicotyledonous plants[93,94].

### De novo assembly of *S. divinorum* transcriptomic data
Illumina RNA-seq reads of *S. divinorum* peltate trichome data were downloaded from ENA Browser (SRR3716680[29], SRR15247010[21], SRR3746809[29,33], and SRR3746810[29,33]) and assembled into transcripts with Trinity. The FPKM values were derived using Hisat2, samtools, and StringTie.

### Cloning of genes
Leaves from *S. divinorum* were collected for RNA extraction using the CTAB method. The extracted RNA was purified with the TURBO DNA-free kit (Thermo Fisher) to eliminate gDNA before reverse transcription into cDNA using the SuperScript IV First-Strand Synthesis System (Thermo Fisher). Candidate genes were amplified with gene-specific primers (Supplementary Table 9) using KOD polymerase (Toyobo). The primers were designed based on genomic and transcriptomic information. Recombinant vectors pTRBO and pEAQ containing the candidate genes were constructed through In-Fusion cloning with ClonExpress II One Step Cloning (Vazyme). After colony PCR, the correctly sequenced gene vectors were utilized for further gene expression.

### Expression of genes in *N. benthamiana*
Recombinant constructs of pTRBO and pEAQ vectors containing either full-length or truncated coding sequences of genes (*AtHMGR*, *TccytGGPPS*, *SspKPS*, *SspKLS*, *SdANS*, *SdHDAS*, *SdDAS*, and *SdCMT*) from

*A. thaliana*, *Taxus canadensis*, *S. splendens*, and *S. divinorum* were introduced into *Agrobacterium tumefaciens* strain GV3101 (Weidi Bio-technology, Shanghai) and then infiltrated into *N. benthamiana*. Positive colonies of transformed *Agrobacterium tumefaciens* were identified by PCR and subsequently cultured in 10 mL LB medium with rifampicin, gentamycin, and kanamycin for 2 days at 200 rpm and 28 °C. After centrifugation at 4000 rpm for 15 min, the cells were resuspended in 1× MMA (10 mM MgCl$_2$, 10 mM MES, and 150 mM acetosyringone) and adjusted to an OD600 of 0.2, then kept in the dark for at least 1 h. *Agrobacterium* strains containing pTRBO or pEAQ plasmids were mixed for co-infiltration into leaves of *N. benthamiana*. The *Agrobacterium* suspensions were then infiltrated into the leaves of 4–6-week-old healthy *N. benthamiana* plants. For the SdCMT products, SAM (*S*-adenosyl methionine) was infiltrated into the plants on day six. On day seven, the infiltrated leaves were collected, ground with a pestle and mortar, and extracted with ethyl acetate overnight at 28 °C while shaking at 220 rpm. The samples were centrifuged, and the supernatant was transferred to a new glass tube, dried using a Rotary Evaporator, and resuspended in 1 mL methanol, with around 100 μL aliquoted for LC-MS analysis.

## LC-MS analyses of *N. benthamiana* extracts

The ethyl acetate extracts of *N. benthamiana* were concentrated using a rotary evaporator, then dissolved in methanol for LC-MS analysis (SCIEX Zeno TOF 7600, SCIEX) on a Kinetex 2.6 μm C-18 100 Å column (100 × 2.1 mm). The aqueous phase (Phase A) consisted of 2 mM ammonium formate with 0.01% formic acid, while the organic phase (Phase B) was acetonitrile. The gradient conditions were as follows: increase Phase B from 0 to 10% over 0 to 0.5 min, from 10 to 40% from 0.5 to 1 min, then ramp up to 99% from 1 to 8 min, holding at 99% for an additional 2 min. Despite the quality of our LC-MS analyses we have not been able to isolate any of the products from *N. benthamiana*. The compounds are unstable and oxidize easily during purification.

During confirmation of divinatorin A formation, we isolated divinatorin A from *Dodonaea viscosa* following previous published methodology[58]. Dried leaves were extracted with 90% EtOH, and the crude extract was partitioned between EtOAc and water. The EtOAc fraction was subjected to reversed-phase column chromatography (EtOH–H$_2$O stepwise gradient). The 85:15 EtOH/H$_2$O fraction was further separated on silica gel using a petroleum ether–EtOAc gradient; the 15:1 petroleum ether/EtOAc subfraction was purified on silica gel with petroleum ether–CH$_2$Cl$_2$ (1:1). Final purification was achieved by reversed-phase chromatography (MeOH–H$_2$O gradient) and ODS gel chromatography (stepwise MeOH–H$_2$O), affording divinatorin A. Isolation was guided by comparison with a small amount of authentic sample provided by Prof. Ai-Jun Hou's laboratory.

Based on the comparison of the LC-MS data of this isolated standard of divinatorin A and the product of the CYP728D25 assays, we were able to confirm the formation of divinatorin A. However, during the sample preparation and analysis of the divinatorin A standard, such oxidations occurred, and some of them were very similar to oxidative products observed in *N. benthamiana* assays (Supplementary Fig. 18).

## Reporting summary

Further information on research design is available in the Nature Portfolio Reporting Summary linked to this article.

## Data availability

The genome assembly and raw sequencing data generated in this study have been deposited in the NCBI GenBank database under BioProject accession PRJNA1284617. The whole-genome assembly is available under accession JBPKIN000000000 [https://www.ncbi.nlm.nih.gov/nuccore/JBPKIN000000000], and raw reads are available in the Sequence Read Archive (SRA) under accessions SRR37143892–SRR37143895 [https://www.ncbi.nlm.nih.gov/sra?

LinkName=bioproject_sra_all&from_uid=1284617]. *S. divinorum* genome annotation data have been uploaded to Figshare [https://doi.org/10.6084/m9.figshare.31524469]. The raw data of LC-MS analysis have been uploaded to Figshare [https://doi.org/10.6084/m9.figshare.31458688] and Science Data Bank [https://doi.org/10.57760/sciencedb.27893]. The sequence data of the reported genes for this study have been deposited in the National Center for Biotechnology Information (NCBI) database (GeneBank accession numbers PX964212-PX964218 and PV857768). Source data are provided with this paper.

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

## Acknowledgements

The authors would like to thank Prof. Ai-Jun Hou (School of Pharmacy, Shanghai Jiao Tong University) for advice and providing resources for divinatorin A. The work was supported by the National Natural Science Foundation of China, Research Fund for International Excellent Young Scientists (RFIS-II) (grant 32150610477, awarded to ECT). HL is supported by the Youth Fund of the National Natural Science Foundation of China (32200313), by Beijing Life Science Academy (BLSA, 2024400CB0050) and by Fundamental Research Funds for the Central Universities. C.M. gratefully acknowledges support from the BBSRC ISP Grant "Harnessing Biosynthesis for Sustainable Food and Health (HBio)" (BB/X01097X/1). C.M. and E.C.T. gratefully acknowledge the Royal Society for the Newton Advanced Fellowship awarded to E.C.T. (NAF⟍R2⟍192001). Y.S. is supported by the Foundation of the Youth Innovation Promotion Association of the Chinese Academy of Sciences.

## Author contributions

E.C.T., H.L., and C.M. initiated and conceived the project and designed the experimental strategy. H.L. assembled the *S. divinorum* genome, transcriptomic data, analysed the genome data with advice from S.W., designed and expressed the synthetic genes from *S. divinorum* in *N. benthamiana*, and analyzed the enzyme assays together with W.X. and B.S. Y.S. reconstituted the pathway on a large scale and isolated and identified the products of SdDAS and SdCMT. C.L. and J.Z. provided resources. H.L., C.M., and E.C.T. wrote the manuscript with valuable input from all authors, who approved the final version of the manuscript.

## Competing interests

The authors declare no competing interests.
