## [Peer Review file · Nature Communications]

Comparative Genome Analysis Provides a Foundation for Defining Salvinorin A Biosynthesis in *Salvia divinorum*.

Corresponding Author: Professor Evangelos Tatsis

Version 0:

Reviewer comments:

Reviewer #1

(Remarks to the Author)

Li et al. investigated the salvinorin A biosynthesis in *Salvia divinorum*. Several steps of this pathway were known before, but this work adds knowledge about two reactions. A genome sequence of *S. divinorum* was generated as part of this study and compared against the existing genome sequences of multiple *Salvia* species. The evolution of the genes/enzymes are explored based on trees and synteny analyses. Please find specific suggestions for improvements below.

Specific comments:

- 1) It is important to differentiate between a genome (DNA) and a genome sequence (information). For example, BUSCO genes are most likely present in the genome, but no corresponding models are detectable in the genome sequence. This needs to be checked throughout the manuscript. To be more concise the authors could also drop "reference" when referring to the genome sequence, because the first genome sequence is always serving as reference for a species until a better one is released. Chromosomes and pseudochromosomes are used interchangeably at some places - please correct.
- 2) Is the traditional knowledge presented in the introduction and its utilization for this study in agreement with the Nagoya protocol?
- 3) Calculating the N50 of pseudochromosomes ("super-N50") does not reveal anything about the genome sequence, because this is mostly determined by the genome structure. Therefore, this should not be presented as a quality indication for the assembly. Instead, it would be interesting to know the N50 at contig level.
- 4) Genome size estimation is tricky and should be performed with multiple tools (GenomeScope2, findGSE, MGSE, gce) to get a range. Also, using a different k-mer size might substantially alter the predicted genome size. This might help to solve the problem that the assembly size (558 Mbp) exceeds the estimated genome size (536 Mbp).
- 5) The authors have used GenomeScope and Smudgeplot using PacBio long sequencing reads, however, the authors clearly state "Currently, GenomeScope and Smudgeplot only support low error short read sequencing" (<https://doi.org/10.1038/s41467-020-14998-3>). Tools that can handle long reads should be considered.
- 6) It would be helpful to mention the type of PacBio reads. Error correction with Illumina data suggests that this must be very old data. The different tool names should not be mentioned in the results section, but in the method section. Line 122-line125 is largely a description of the method. The actual results could be summarized in a few sentences.
- 7) Fig2 CIRCOS plot could use colour scale at the side for density heatmaps. Line 160 ..., 'and genes' should be 'protein-coding genes' (assuming they are protein-coding genes).
- 8) There is not much information about the annotated protein-coding genes besides the number of genes. What percentage of the BUSCO genes could be recovered compared to the assembly BUSCO?
- 9) It appears that the results of the "Comparative Genomic Analysis" are largely only confirming previous studies without

providing new insights. Therefore, it might be better to move this analysis into the supplements to focus on the discovery of new biosynthesis steps.

10) Figure 4: There is no synteny displayed in the comparison of genome sequences. Also, the connected genes are often located at the end of a contig. Please indicate which of the displayed genes are actually syntelogs. To increase the resolution of the displayed region, it might be helpful to use only the closest outgroup (*Sesamum indicum*).

11) Figure 5: Please include the species names in the labels of the tree displayed in a. Bootstrap values on the tree are missing.

12) Figure 7: The crucial gene is located at the end of a contig in two species. The outgroup does not display the relevant locus. A synteny analysis is only meaningful if upstream and downstream flanking genes of the gene of interest are presented. The tree shown in C lacks bootstrap/support values.

13) The number of figures is very high and many of them are not strictly required to communicate the findings. The authors might want to move some of them into the supplements.

14) Line 548-559 reads like another paragraph of the introduction. It might be better to integrate this in the introduction, because it is not directly connected to the results of the present study.

15) The method section contains results (e.g. genome size in line 642).

16) References for the tools and protocols used are largely missing in the method section. It appears that some references are also not properly formatted. Versions and parameters need to be provided for all tools.

17) The correction of the assembly with Illumina reads is not described.

18) What are the four different plots in Figure S1 that are not labeled? Are there replicates?

19) Figure S2: Why not use the latest GenomeScope version? What would be the result for other k-mer sizes? It is well known that the results can heavily depend on the k-mer size.

20) Figure S5: The similarity in this plot is high, but more in-depth comparison for individual chromosome sequences would be helpful to see differences.

21) Figure S6: The comparison of the Hi-C mounting rate does not make sense. I would suggest to restrict it to the other three plots if any. Since they are just displaying simple numbers, the content could also be summarized in a sentence. The important question is why is the number of pseudochromosomes different?

22) Figure S9 looks chaotic. It is not clear what message should be conferred by this figure.

23) Figure S10/11: species names could be written in italics.

24) Figure S12 is hard to interpret. It would be better to include full species names in this tree and not just to refer to Table S5.

25) Supplementary Tables as PDF are broken. Publishing these as separate files/tables might be better.

26) Since the divergence time of the Asian and American *Salvia* clade (21.6MYA) and the estimated WGD α at 18.46 MYA are close, it is also necessary to include Asian *Salvia* species in the Ks distribution analysis (Fig S8) and show that the Ks peak at 0.24 is not shared by the Asian *Salvia* species to confirm that WGD α event occurred after divergence from Asian *Salvia* species.

27) The content of Tables S5 and S6 described in text does not match the actual content. Looks like some tables are missing.

28) Which transcriptome data was used for structural annotation? Is it publicly available? If so, please provide SRA accessions.

29) Method descriptions for constructing the gene trees are missing.

Minor comments:

line126: genomescope > GenomeScope

line 234/235: genera names should be written in italics

line 503: proposed

line 643: DNB should be explained at first occurrence

line 769: Supplemental Table ?

Discrepancy in text and figures at multiple places, eg CYP716AH39 line 80 vs CYP76AH39 in Fig1a; hardwickiic (Fig1a and multiple places in text) vs hardwickic at other places; Fig 5 SspANS vs SpANS in text

Reviewer #2

(Remarks to the Author)

Reviewer #3

(Remarks to the Author)

The article submitted by Li and coworkers focuses on the biosynthesis of salvinorin A, a furanoclerodane diterpenoid found in *Salvia divinorum* and closely related species that has promising possible applications in treating drug addiction, pain, and neuropsychiatric disorders. The authors developed an updated genome of *Salvia divinorum* of advanced chromosome-level resolution and used comparative genomics with related *Salvia* species that do not produce salvinorin A to clarify some of the early pathway reactions. Functional characterization of a predicted P450 and methyl transferase in yeast and *Nicotiana benthamiana* revealed additional P450 functional decoration reactions en route to salvinorin A. Overall, the experiments are well performed and convincing, and the illustrations are clear and provide all required data to support the presented conclusions. Below are a few comments that the authors might find helpful.

Major Comments:

(1) The shown products of CYP728D26 and SdCMT, especially the formation of divinatorin A, is well supported based on the MS/MS results. However, it is not clear how the position of functional groups and especially the stereochemistry of the molecules were determined. Seeing that the purity of the products in yeast and more so in *N. benthamiana* is quite good, if the compounds can be produced in sufficient abundance, additional NMR structural verification would substantially strengthen the manuscript.

(2) The data provided for possible additional, trichome-specific, P450 enzymes acting in salvinorin A biosynthesis is convincing. As such, for this level of publication, the fact that none of these candidates were tested leaves the reader somewhat wanting. The authors may want to publish this work in a later manuscript. However, in favor of the completeness of the presented study, at least a functional testing of some of the candidates would strengthen this manuscript. Alternatively, the authors may consider removing this predictive part of the results.

(3) Much of the discussion read more like a recap of the results. The authors could consider providing a broader discussion including, for example, the evolution of furanoclerodanes within and beyond Lamiaceae, as well as the neofunctionalization of pathway genes subsequent to the whole genome duplication. For example, prior work identified active site residues, modification of which has likely contributed to the evolution of the kolavenyl diphosphate synthase activity that catalyzes the first committed step (e.g., Chen et al., 2017 *J Exp Bot.* 68:1109-1122; Pelot et al., 2017 *Plant J.* 89:885-897).

Minor Comments:

(4) The abstract focuses a lot on future experiments and promises. A more detailed description of some of the key findings such as the P450 enzymes would be helpful.

(5) Line 79: To my knowledge it is not clear if plastidial diterpene intermediates are actively exported or transferred, for example, via hemifusion of membranes.

(6) Line 89: I assume the red area in Figure 1 (map) indicates the *S. divinorum* distribution? This is not indicated in the figure legend.

(7) Line 96: There are several different enzyme names for the early steps of salvinorin A biosynthesis in the current literature. As this manuscript provides an updated genome, it provides a good opportunity to consolidate the enzymes IDs. Maybe the authors want to consider doing so in Figure 1 by listing the published IDs and defining those to move forward with.

(8) There are numerous typos and formatting errors throughout the manuscript. I recommend that the authors carefully revise the manuscript. Some examples are:

- Throughout the text, there is a mixed use of British and American English.
- Line 219: typo? T(SRR...
- Line 238: abbreviation of MYA should be introduced already in line 179.
- Line 240: Typo – should be 'Lamiaceae'

- Line 320: Typo – space in CYP728D26
- Line 323: 'S. splendens' needs to be italicized
- Line 343: Typo – should be 'tumefaciens'
- Line 344: 'the' should not be italicized
- Line 503: Typo – proposed not 'froposed'

Reviewer #4

(Remarks to the Author)

This is a review of the article entitled "Comparative Genome Analysis Provides a Foundation for the Elucidation of Salvinatorin A Biosynthesis in *Salvia divinorum*" by Li and coworkers. The authors present a detailed comparative genomic analysis and characterization of two key activities in the biosynthesis of divinatorins (apparent precursors to salvatorins). The study is interesting and, while it would have been amazing for the manuscript to report a completed pathway to salvinatorin A, it does indeed 'provide a foundation for the elucidation of Salvinatorin A biosynthesis'. Below are several suggestions for improving the manuscript.

Abstract, lines 27-30: The final sentence of the abstract could be split into two sentences.

Introduction, line 85: Comma missing between reference number 21 and 31 in superscript

Results, line 219: "(SRR15247010)" is the "t" before the first parenthesis a typo? What tissue does this refer to?

Results, line 240: Lamiaceae spelling?

Results, line 310: "forma" typo?

Results, line 317-319: It is somewhat unclear what is meant by "Unfortunately, due to deficiencies in the yeast assay system, the relevant enzyme products were mistakenly identified, although the catalytic activities of the enzymes on the clerodane scaffold were identified correctly." → how were the enzyme activities identified correctly if the enzyme products were mistakenly identified? Perhaps I am misunderstanding something about this statement. I recommend rewording to make it more clear how the enzyme activities were confirmed despite the problems with the yeast assay system.

Results, line 326: "CYP7" typo? Should this be "CYB728D27"?

Results, line 331: "Since CYP728D26 has been reported to oxidize C-1 of clerodane scaffold and install a hydroxyl group", does this need a citation?

Results, line 333/334: Consider using the same abbreviation for tandem mass spec for consistency (MS2 or MS/MS)

Results, line 362: "wit" -> "with"

Results, line 328 it is stated that only CYP728D25 (not CYP728D27) displayed activity toward hardwickiic acid, but then on lines 366-367 it is stated that CYP728D27 might contribute to the hydroxylations that were observed in the *N. benthamiana* assays when CYP728D25 was co-infiltrated with the enzymes required for making the hardwickiic acid scaffold. Were *N. benthamiana* assays performed with CYP728D27? If not, it might be appropriate to tone down the sentence in question as there is not strong evidence supporting the hydroxylation of C1, C2, and C17 of the hardwickiic acid scaffold by CYP728D27 based on the data presented. Alternatively, please present the data for CYP28D27

Results, line 387: "thechemical"

Results, line 430: "fonthas"

Results, line 438-440: Wording of this sentence is confusing. Consider rewording to improve clarity.

Discussion, lines 618-619: Are these results based on data collected by the authors of this paper or the literature? Please provide supporting information or reference.

If other genomes versions are less complete, what does that mean for comparative genomics?
label subgenera in Fig 3A

The statement "activity of annonene synthase, in both *S. divinorum* and *S. splendens*, has evolved by recruitment of genes encoding cytochrome P450 enzymes originally with ferruginol synthase activity and neofunctionalization towards annonene synthase activity" needs to be supported with a citation or Fig. 5 needs to be updated to show which tips correspond to enzymes with FS activity.

Regarding the statement "Our data suggest that in *N. benthamiana*, SdHDAS (CYP728D26) can synthesise some divinatorin A and its isomer" perhaps I am missing something, but results from heterologous expression of CYP728D26 should be shown, perhaps as a supplemental figure. Or perhaps this information is coming from the previous study alluded

to on line 331? Overall, there seems to be some confusion or inconsistency regarding the various P450s. Please update the manuscript to make it absolutely clear which groups did which work and when, as well as which enzymes have which activities, and present all relevant data.

A number of paragraphs in the discussion are redundant with the results section and thus do relatively little to enrich the manuscript. I suggest that the discussion either be condensed, merged with the results section to create a combined section with less redundancy, or be expanded to touch on, for example, an analysis of comparative genomics as applied pathway discovery - what have such approaches, like that shown here, in combination with those preceding it, taught us overall? What are some new directions in which we could go using comparative genomics to study pathways at scale?

Version 1:

Reviewer comments:

Reviewer #3

(Remarks to the Author)

In this revised version of the manuscript, the authors have satisfactorily addressed my comments on the original submission. I appreciate the effort in biochemically characterizing additional P450 candidates to provide more expansive insight into the broader diterpene pathway network. I appreciate the issues in purifying some of the enzyme products due to their low abundance and instability. Given these circumstances, I think the provided LC-MS based product identification offers sufficient evidence. In my view, this article makes a timely and valuable contribution to the broader field of plant metabolism and plant natural products.

Reviewer #4

(Remarks to the Author)

The authors have satisfactorily addressed my comments and I recommend the article for publication.

Reviewer #5

(Remarks to the Author)

Dear authors.

The revised manuscript has been carefully revised in accordance with the reviewer's comments. In addition, four minor comments have been added, and we would appreciate your kind review.

- Figure 4 clearly demonstrates that class I clerodane synthases (KSL2) are syntenically conserved only within *Salvia*, while being absent from related Lamiaceae lineages such as *Sesamum*, *Callicarpa*, and *Scutellaria*. A brief clarification of the biosynthetic implications of this absence (i.e., whether class II activity alone is insufficient for salvinatorin-type diterpene formation) would help guide the reader's interpretation of this figure.

- Figure 5 provides compelling phylogenetic and syntenic evidence that annonene synthase activity in American *Salvia* species arose via neofunctionalization of CYP76AH enzymes originally associated with ferruginol synthase activity. In addition, the identification of a shared WGD α event among these species suggests a plausible genomic context for this functional divergence.

To further improve clarity for the reader, the authors may consider adding a brief sentence explicitly linking the inferred neofunctionalization of CYP76AH enzymes to the WGD α event (e.g., whether one duplicated copy retained ferruginol synthase activity while the other acquired annonene synthase activity). This clarification would strengthen the interpretation of Figure 5 without altering the overall conclusions of the study.

- Ensure consistent spelling of hardwickiic acid throughout the manuscript (text and figures), as alternative spellings (e.g. harwickiic) appear in some places.

- In the description of Figure 5b, the term "nonfunctionalized CYP76AH enzymes" is used. Could the authors please clarify whether this term was intended to convey a specific biological meaning, or whether "neofunctionalized" was intended instead?

Sincerely,

In summary, for this revised manuscript:

We have addressed Reviewer 1's comments by re-estimating genome size using multiple tools (GenomeScope2, findGSE, MGSE, gce) and different k-mer sizes. GCE results predicted a genome size of 553 Mbp and 556 Mbp for k-mer sizes 17 and 19, respectively. When using GenomeScope and GenomeScope 2 the estimated genome size ranged from 550 Mbp to 552 Mbp depending on k-mer size. This newly estimated genome size is closer to the assembly size and these results are shown in Table S2. To provide more direct assessment, we have also performed flow cytometry to evaluate the genome size, experimentally (Fig. S1). We used plotsr to provide a more detailed comparison of individual chromosome sequences, as shown in Figure S5. A new Ks distribution analysis has been provided. We recalculated the Ks and the results showed that the Ks peak at 0.24 is not shared by the Asian *Salvia* species (Fig. S8).

In addition to address the comments of reviewers 3 and 4, we used the new genome sequence to establish three new enzyme activities core to the production of divinorin intermediates and showed how these activities evolved through gene duplication and subfunctionalisation (Figs 7 and 9). We were able to use a divinorin A standard to prove that one of the products of CYP728D25 was divinorin A. This standard also allowed us to identify the susceptibility of divinorins to oxidation, an instability that has likely been a major bottleneck in establishing the biosynthetic pathway for salvinatorin A. In assays in *N. benthamiana*, we showed that CYP728D27 is active on hardwickiic acid as a hydroxylase producing isomers of divinorin A but with different hydroxylation positions to CYP728D25, and that CYP728D25 has a second hydroxylase activity, probably on C-17. We propose, for the first time, that a network of hydroxylations underpins the diversity of divinorins produced by *S. divinorum*. We performed a new syntenic analysis with the extended syntenic area for Figure 7 and added this extended syntenic analysis to the SI. We have added bootstrap values >50 to Fig 7 Panel C.

Finally, the Discussion section has been completely rewritten. We have added three new authors who have contributed significantly to the revision.

Below, we provide a detailed, point-by-point responses to all comments, and indicate where changes have been made in the revised manuscript.

Reviewer #1 (Remarks to the Author):

Li et al. investigated the salvinatorin A biosynthesis in *Salvia divinorum*. Several steps of this pathway were known before, but this work adds knowledge about two reactions. A genome sequence of *S. divinorum* was generated as part of this study and compared against the existing genome sequences of multiple *Salvia* species. The evolution of the genes/enzymes are explored based on trees and synteny analyses. Please find specific suggestions for improvements below.

We would like to thank Reviewer #1 for valuable insight into genome assembly quality, methodology, and figure presentation, issues which we have addressed through substantial revision

Specific comments:

1) It is important to differentiate between a genome (DNA) and a genome sequence (information). For example, BUSCO genes are most likely present in the genome, but no corresponding models are detectable in the genome sequence. This needs to be checked throughout the manuscript. To be more concise the authors could also drop "reference" when referring to the genome sequence, because the first genome sequence is always serving as reference for a species until a better one is released. Chromosomes and pseudochromosomes are used interchangeably at some places - please correct.

We revised the manuscript to use "genome sequence" consistently for sequence information and "genome" only when referring to the physical entity. Redundant use of "reference genome sequence" has been removed, and terminology distinguishing "chromosomes" from "pseudochromosomes" has been introduced throughout.

2) Is the traditional knowledge presented in the introduction and its utilization for this study in agreement with the Nagoya protocol?

We clarified in the Methods section that all plant material was obtained in accordance with institutional guidelines and international conventions, including the Nagoya Protocol.

3) Calculating the N50 of pseudochromosomes ("super-N50") does not reveal anything about the genome sequence, because this is mostly determined by the genome structure. Therefore, this should not be presented as a quality indication for the assembly. Instead, it would be interesting to know the N50 at contig level.

We removed pseudochromosome "super-N50" values and now report contig-level N50 values as a more meaningful measure of assembly quality (Results, Table S4). Accidentally, we wrote that we used Illumina reads in the original submission to correct the contigs – the manuscript was prepared in a hurry to meet the impending submission deadline. We apologize for this error. Since the HiFi reads were already accurate there was no need for such correction of the output from the Hifiasm assembly pipeline.

Cheng, H., Concepcion, G.T., Feng, X., Zhang, H., Li H. (2021) Haplotype-resolved de novo assembly using phased assembly graphs with hifiasm. *Nat Methods*, 18:170-175.
<https://doi.org/10.1038/s41592-020-01056-5>

Cheng, H., Jarvis, E.D., Fedrigo, O., Koepfli, K.P., Urban, L., Gemmell, N.J., Li, H. (2022) Haplotype-resolved assembly of diploid genomes without parental data. *Nature Biotechnology*, 40:1332–1335.
<https://doi.org/10.1038/s41587-022-01261-x>

4) Genome size estimation is tricky and should be performed with multiple tools (GenomeScope2, findGSE, MGSE, gce) to get a range. Also, using a different k-mer size might substantially alter the predicted genome size. This might help to solve the problem that the assembly size (558 Mbp) exceeds the estimated genome size (536 Mbp).

As suggested, we have re-estimated genome size using multiple tools (GenomeScope2, findGSE, MGSE, gce) and different k-mer sizes. GCE results predicted a genome size of 553 Mbp and 556 Mbp for k-mer sizes 17 and 19, respectively. While using GenomeScope and GenomeScope 2 the estimated genome size ranged from 550 Mbp to 552 Mbp depending on k-mer size. This newly estimated genome size is closer to assembly size. These results are shown in Table S2, providing a range that contextualizes the assembly size relative to estimated genome size.

5) The authors have used GenomeScope and Smudgeplot using PacBio long sequencing reads, however, the authors clearly state "Currently, GenomeScope and Smudgeplot only support low error short read sequencing" (<https://doi.org/10.1038/s41467-020-14998-3>). Tools that can handle long reads should be considered.

The cited sentence is from the 2020 publication of GenomeScope 2.0 (Ranallo-Benavidez et al., 2020), which, indeed, warned against the use of high-error single-molecule sequencing data such as PacBio CLR or Oxford Nanopore. However, at that time, high-accuracy long-read sequencing technologies such as PacBio HiFi were not widely available.

Since then, PacBio HiFi sequencing has become established as a highly accurate platform, providing Q30 or higher read quality (>99.9%), comparable to Illumina short reads. These HiFi reads produce clean k-mer frequency profiles without the noise associated with older long-read technologies, making them well-suited for tools like GenomeScope and Smudgeplot. In fact, the GenomeScope GitHub repository now includes numerous community discussions and successful use cases with HiFi data (e.g., <https://github.com/schatzlab/genomescope/issues>).

We agree that genome size estimations based on k-mer frequencies can vary depending on genomic features such as heterozygosity, repeat content, or allelic duplication, and that both short- and long-read-based methods may be affected. Thus, genome size estimates from bioinformatic tools usually serve as approximations rather than exact measurements.

To provide more direct assessment, we have also performed flow cytometry to evaluate the genome size, experimentally (Fig. S1)

6) It would be helpful to mention the type of PacBio reads. Error correction with Illumina data suggests that this must be very old data. The different tool names should not be mentioned in the results section, but in the method section. Line 122-line125 is largely a description of the method. The actual results could be summarized in a few sentences.

Please refer to the answer to Reviewer 1, point 5 above.

7) Fig2 CIRCOS plot could use colour scale at the side for density heatmaps. Line 160 ..., 'and genes' should be 'protein-coding genes' (assuming they are protein-coding genes).

The title protein-coding genes has been added

8) There is not much information about the annotated protein-coding genes besides the number of genes. What percentage of the BUSCO genes could be recovered compared to the assembly BUSCO?

We analyzed the BUSCO in protein mode and we found 99.2% of BUSCO genes are complete. We now provide the BUSCO analysis in protein mode in Table S4b.

9) It appears that the results of the "Comparative Genomic Analysis" are largely only confirming previous studies without providing new insights. Therefore, it might be better to move this analysis into the supplements to focus on the discovery of new biosynthesis steps.

We disagree that the new genome provides little by way of new insight compared to the previous genome sequence. Beyond identifying the genes catalysing the correct biosynthetic pathway to hardwickiic acid (including the class I diterpene synthases and the recruitment of genes encoding both diterpene synthases and CYP450 enzymes from other diterpene synthetic pathways in the Calosphace subgenus of *Salvia*), the phylogenetic analyses (Figs 4 and 5) support further the polyphyletic origin of clerodane metabolism in Lamiaceae, especially the *Salvia*-specific emergence of class I clerodane synthases and show that neofunctionalization of annonene synthase was a result of a WGD event specific to neotropical *Salvia* species. In addition, we used the new genome sequence to establish three new enzyme activities core to the production of divinorin intermediates and showed how these activities evolved through gene duplication and subfunctionalisation (Figs 7 and 9). We believe that these advances demonstrate well the important role that comparative genomics can play in understanding complex pathways in the specialized metabolism of medicinal plants, how these specialized pathways evolved and, most importantly, how they might be harnessed in the future for therapeutic applications.

10) Figure 4: There is no synteny displayed in the comparison of genome sequences. Also, the connected genes are often located at the end of a contig. Please indicate which of the displayed genes are actually syntelogs. To increase the resolution of the displayed region, it might be helpful to use only the closest outgroup (*Sesamum indicum*).

The resolution of Figure 4 has been improved and the analysis has been re-run to include *Sesamum indicum* as an outgroup.

11) Figure 5: Please include the species names in the labels of the tree displayed in a. Bootstrap values on the tree are missing.

The species name and bootstrap values over 50 have been added.

12) Figure 7: The crucial gene is located at the end of a contig in two species. The outgroup does not display the relevant locus. A synteny analysis is only meaningful if upstream and downstream flanking genes of the gene of interest are presented. The tree shown in C lacks bootstrap/support values.

We performed a new syntenic analysis with extended syntenic area but the new synteny did not reflect clearly the syntenic relationships any more clearly than in the original version. Therefore we added this extended syntenic analysis in SI (Fig S15). We have added bootstrap values >50 to Fig 7 Panel C.

13) The number of figures is very high and many of them are not strictly required to communicate the findings. The authors might want to move some of them into the supplements.

We have removed figure 10.

14) Line 548-559 reads like another paragraph of the introduction. It might be better to integrate this in the introduction, because it is not directly connected to the results of the present study.

The Discussion section has been completely rewritten.

15) References for the tools and protocols used are largely missing in the method section. It appears that some references are also not properly formatted. Versions and parameters need to be provided for all tools.

References have been formatted appropriately using EndNote.

16) The correction of the assembly with Illumina reads is not described.

As explained in the responses to points 5 and 6 the statement about correction with Illumina reads was entered in error in the original submission and has now been removed.

17) What are the four different plots in Figure S1 that are not labeled? Are there replicates?

The left column shows three replicates of *Solanum lycopersicum*, and the right column shows three replicates of *Salvia divinorum* as described in the legend.

18) Figure S2: Why not use the latest GenomeScope version? What would be the result for other k-mer sizes? It is well known that the results can heavily depend on the k-mer size.

A new version of Figure S2 has been provided using the latest versions of both GenomeScope (v1.0.0) and GenomeScope2 (v2.0.1), and the predicted genome size with different k-mer sizes, which we have included in the results in SI (Fig. S2 and Table S2).

19) Figure S5: The similarity in this plot is high, but more in-depth comparison for individual chromosome sequences would be helpful to see differences.

We have now used plotsr to provide a detailed comparison of individual chromosome sequences, as shown in Figure S5.

20) Figure S6: The comparison of the Hi-C mounting rate does not make sense. I would suggest to restrict it to the other three plots if any. Since they are just displaying simple numbers, the content could also be summarized in a sentence. The important question is why is the number of pseudochromosomes different?

The reviewer seems to have misunderstood what is shown in Figure S6 which is a simple comparison between the first *S. divinorum* genome (published by Ford et al., 2024) and our new one. The reason why the haploid number of chromosomes in our genome is lower than in Ford et al., is that we were able to bridge the gaps in their assembly (see Supplementary Figure S5). Our conclusion of a haploid genome of n=11 fits with cytogenetic estimates (Casselmann et al., 2014).

21) Figure S9 looks chaotic. It is not clear what message should be conferred by this figure.

We have changed the color to make the figure easier to understand and added a sentence to the legend for Figure S9: Conserved syntenic regions between chromosomes of *Salvia divinorum*, *Salvia hispanica* and *Salvia splendens* are highlighted with different colors.

22) Figure S12 is hard to interpret. It would be better to include full species names in this tree and not just to refer to Table S5.

We were unable to include the full species names in the updated Figure S16 but we have added them to the legend below.

23) Supplementary Tables as PDF are broken. Publishing these as separate files/tables might be better.

Thanks for the suggestion.

24) Since the divergence time of the Asian and American *Salvia* clade (21.6MYA) and the estimated WGD α at 18.46 MYA are close, it is also necessary to include Asian *Salvia* species in the Ks distribution analysis (Fig S8) and show that the Ks peak at 0.24 is not shared by the Asian *Salvia* species to confirm that WGD α event occurred after divergence from Asian *Salvia* species.

A new Ks distribution analysis has been provided (Figure S8). We recalculated the Ks and the results showed that the Ks peak at 0.24 is not shared by the Asian *Salvia* species.

25) The content of Tables S5 and S6 described in text does not match the actual content. Looks like some tables are missing.

Corrected

26) Which transcriptome data was used for structural annotation? Is it publicly available? If so, please provide SRA accessions.

We provided the SRA accessions in the Materials & Methods section, Lines 754-756: Illumina RNA-seq reads of *S. divinatorum* peltate trichome data were downloaded from ENA Browser (SRR3716680, SRR15247010, SRR3746809, SRR3746810) and assembled into transcripts using Trinity.

27) Method descriptions for constructing the gene trees are missing.

We have provided the following method descriptions:

Phylogeny of CYP450 and CMT genes

CYP450 protein sequences were chosen based on syntenic analysis, while CMTs were selected from the literature. Protein sequences were aligned using ClustalW in MEGA12. The best-fit substitution models were JTT + I + G for CYP76AH, JTT + I + G + F for CYP728D, and LG + G + I for CMTs, with a bootstrap value of 1000. Phylogenetic trees were constructed using the Maximum Likelihood method. These descriptions can be found in Lines 742-747 of the Materials & Methods section.

Minor comments:

line126: genomescope > GenomeScope

Corrected, also on Line 709.

line 234/235: *genera* names should be written in italics

Corrected.

line 503: proposed

Corrected

line 643: DNB should be explained at first occurrence

Done.

line 769: Supplemental Table ?

Corrected Discrepancy in text and figures at multiple places,

eg CYP716AH39 line 80 vs CYP76AH39 in Fig1a;

Corrected

hardwickiic (Fig1a and multiple places in text) vs hardwickic at other places;

Corrected to hardwickiic.

Fig 5 SspANS vs SpANS in text

Corrected to SspANS throughout the manuscript.

Reviewer #2 (Remarks to the Author):

We thank Reviewer #2 for their participation in the review process.

Reviewer #3 (Remarks to the Author):

The article submitted by Li and coworkers focuses on the biosynthesis of salvinatorin A, a furanoclerodane diterpenoid found in *Salvia divinorum* and closely related species that has promising possible applications in treating drug addiction, pain, and neuropsychiatric disorders. The authors developed an updated genome of *Salvia divinorum* of advanced chromosome-level resolution and used comparative genomics with related *Salvia* species that do not produce salvinatorin A to clarify some of the early pathway reactions. Functional characterization of a predicted P450 and methyl transferase in yeast and *Nicotiana benthamiana* revealed additional P450 functional decoration reactions en route to salvinatorin A. Overall, the experiments are well performed and convincing, and the illustrations are clear and provide all required data to support the presented conclusions. Below are a few comments that the authors might find helpful.

We thank Reviewer #3 for their comments, and suggestions to emphasize the biochemical validation and broaden the discussion, suggestions which we have incorporated into the revised manuscript to strengthen the impact of the manuscript.

Major Comments:

(1) The shown products of CYP728D26 and SdCMT, especially the formation of divinatorin A, is well supported based on the MS/MS results. However, it is not clear how the position of functional groups and especially the stereochemistry of the molecules were determined. Seeing that the purity of the products in yeast and more so in *N. benthamiana* is quite good, if the compounds can be produced in sufficient abundance, additional NMR structural verification would substantially strengthen the manuscript.

Despite the quality of our LC-MS analyses we have not been able to isolate any of the products from *N. benthamiana*. The compounds are unstable and oxidize easily during purification. During confirmation of divinatorin A formation, we isolated divinatorin A from *Dodonaea viscosa* extract as described by Zhao et al *Bioorganic Chemistry*, 112, 104916 (2021). Based on comparison of the LC-MS data of this isolated standard of divinatorin A and the product of the CYP728D25 assays we were

able to confirm the formation of divinatorin A. However, during the sample preparation and analysis of the divinatorin A standard, such oxidations occurred, and some of them were very similar to oxidative products observed in *N. benthamiana* assays.

Divinatorin A (standard)

AtHMGR+TcGGPPS+SspKPS+SspKLS+SdANS+SdHDAS+CYP728D25(SdDAS)

First row: Total Ion Chromatogram (TIC) of isolated divinatorin A standard.

Second row: Extracted Ion Chromatograms at m/z 331.192, 347.187, and 363.183 of isolated divinatorin A sample.

Third row: Extracted Ion Chromatograms at m/z 331.192, 347.187, and 363.183 from assays of CYP728D25 in *N. benthamiana*.

This susceptibility to oxidation means that divinatorin A and its derivatives are very difficult to purify from *N. benthamiana* and further chemical confirmation of intermediates will need to await the development of better production systems, possibly in alternative heterologous hosts.

(2) The data provided for possible additional, trichome-specific, P450 enzymes acting in salvinatorin A biosynthesis is convincing. As such, for this level of publication, the fact that none of these candidates were tested leaves the reader somewhat wanting. The authors may want to publish this work in a later manuscript. However, in favor of the completeness of the presented study, at least a functional testing of some of the candidates would strengthen this manuscript. Alternatively, the authors may consider removing this predictive part of the results.

Following this suggestion we expanded our work to assaying CYP728D27 which is also highly expressed in trichomes of *S. divinorum*. In assays in *N. benthamiana*, we show that CYP728D27 is active on hardwickiic acid as a hydroxylase producing isomers of divinatorin A but with different hydroxylation positions to CYP728D25, and that CYP728D25 has a second hydroxylase activity, probably on C-17. Consequently, we followed the advice of R3 and decided to remove completely the part entitled "The missing steps in biosynthesis of salvinatorin A." where potential candidate P450 enzymes were described, and replace this with a more detailed discussion of the network of hydroxylations that underpin the diversity of divinatorins produced by *S. divinorum*. It is possible that

this same network also underpins much of the diversity of salviarins in *S. divinorum* and so redefines the number of new genes/enzymes necessary for the synthesis of salvinorin A.

(3) Much of the discussion read more like a recap of the results. The authors could consider providing a broader discussion including, for example, the evolution of furanoclerodanes within and beyond Lamiaceae, as well as the neofunctionalization of pathway genes subsequent to the whole genome duplication. For example, prior work identified active site residues, modification of which has likely contributed to the evolution of the kolavenyl diphosphate synthase activity that catalyzes the first committed step (e.g., Chen et al., 2017 J Exp Bot. 68:1109-1122; Pelot et al., 2017 Plant J. 89:885-897).

We have addressed this suggestion by enriching the discussion by reference to previous work on SdKPS activity.

Minor Comments:

(4) The abstract focuses a lot on future experiments and promises. A more detailed description of some of the key findings such as the P450 enzymes would be helpful.

We have updated the Abstract to emphasise better our findings and to provide more balance to future promise.

(5) Line 79: To my knowledge it is not clear if plastidial diterpene intermediates are actively exported or transferred, for example, via hemifusion of membranes.

We agree and we have reworded this part as follows:

“It is not clear if diterpenes are actively exported³⁰ or transferred via hemifusion at contact sites between plastid and ER^{31,32} for later oxidations by cytochrome P450 enzymes”.

(6) Line 89: I assume the red area in Figure 1 (map) indicates the *S. divinorum* distribution? This is not indicated in the figure legend.

We have included a description of the distribution of *S. divinorum* in the figure legend.

(7) Line 96: There are several different enzyme names for the early steps of salvinorin A biosynthesis in the current literature. As this manuscript provides an updated genome, it provides a good opportunity to consolidate the enzymes IDs. Maybe the authors want to consider doing so in Figure 1 by listing the published IDs and defining those to move forward with.

We understand the concern of the reviewer regarding enzyme nomenclature. We believe it is necessary to include enzyme names for both *Salvia splendens* (in green) and *Salvia divinorum* enzymes (in purple) to assist the reader in navigating this difficult terminology, especially in the later evolutionary analysis.

(8) There are numerous typos and formatting errors throughout the manuscript. I recommend that the authors careful revise the manuscript. Some examples are:

Line 219: typo? T(SRR...

Corrected

Line 238: abbreviation of MYA should be introduced already in line 179.

Corrected

Line 240: Typo – should be ‘Lamiaceae’

Corrected

Line 320: Typo – space in CYP728D26

Corrected

Line 323: ‘*S. splendens*’ needs to be italicized

Corrected

Line 343: Typo – should be ‘tumefaciens’

Corrected

Line 344: ‘the’ should not be italicized

Corrected

Line 503: Typo – proposed not ‘froposed’

This was removed

Reviewer #4 (Remarks to the Author):

This is a review of the article entitled “Comparative Genome Analysis Provides a Foundation for the Elucidation of Salvinorin A Biosynthesis in *Salvia divinorum*” by Li and coworkers. The authors present a detailed comparative genomic analysis and characterization of two key activities in the biosynthesis of divinorins (apparent precursors to salvatorins). The study is interesting and, while it would have been amazing for the manuscript to report a completed pathway to salvinorin A, it does indeed ‘provide a foundation for the elucidation of Salvinorin A biosynthesis’. Below are several suggestions for improving the manuscript.

We would like to thank Reviewer #4 for highlighting necessary clarifications and consistency issues, which we believe we have resolved and which have improved the readability and accuracy of the manuscript.

Abstract, lines 27-30: The final sentence of the abstract could be split into two sentences.

The final sentence of the abstract is now split as follows:

“Our genome-based, microevolutionary approach provides insight into how specialized furanoclerodanes of immense pharmacological importance evolved. The results lay a clear path for identification of the remaining steps in the biosynthetic pathway which could lead to synthetic production for the development of new therapeutics”.

Introduction, line 85: Comma missing between reference number 21 and 31 in superscript

Corrected

Results, line 219: “t(SRR15247010)” is the “t” before the first parenthesis a typo? What tissue does this refer to?

Typo corrected

Results, line 240: Lamiaceae spelling?

Corrected

Results, line 310: “forma” typo?

Typo corrected

Results, line 317-319: It is somewhat unclear what is meant by “Unfortunately, due to deficiencies in the yeast assay system, the relevant enzyme products were mistakenly identified, although the catalytic activities of the enzymes on the clerodane scaffold were identified correctly.” → how were the enzyme activities identified correctly if the enzyme products were mistakenly identified? Perhaps I am misunderstanding something about this statement. I recommend rewording to make it more clear how the enzyme activities were confirmed despite the problems with the yeast assay system.

We decided to remove this part to avoid any confusion

Results, line 326: “CYP7” typo? Should this be “CYB728D27”?

Yes it should be CYP728D27. Corrected.

Results, line 331: "Since CYP728D26 has been reported to oxidize C-1 of clerodane scaffold and install a hydroxyl group", does this need a citation?

Added.

Results, line 333/334: Consider using the same abbreviation for tandem mass spec for consistency (MS2 or MS/MS)

Noted and corrected.

Results, line 362: "wit" -> "with"

Corrected

Results, line 328 it is stated that only CYP728D25 (not CYP728D27) displayed activity toward hardwickiic acid, but then on lines 366-367 it is stated that CYP728D27 might contribute to the hydroxylations that were observed in the *N. benthamiana* assays when CYP728D25 was co-infiltrated with the enzymes required for making the hardwickiic acid scaffold. Were *N. benthamiana* assays performed with CYP728D27? If not, it might be appropriate to tone down the sentence in question as there is not strong evidence supporting the hydroxylation of C1, C2, and C17 of the hardwickiic acid scaffold by CYP728D27 based on the data presented. Alternatively, please present the data for CYP28D27

We followed the suggestion of reviewer and we have assayed CYP728D27 towards hardwickiic acid and in combination with CYP728D25. CYP728D27 is active on hardwickiic acid acting as hydroxylase. Our data show that it can catalyse two hydroxylations of the furanoclerodane scaffold, as can CYP726D25, but that the positions of these hydroxylations are different.

We tried to isolate the products of CYP728D25, CYP728D27 and SdCMT, however we were not able to isolate any of the products. As reported in the response to Reviewer 3 (1) we believe this problem to be due to the susceptibility of divinorins to oxidation especially during LC-MS analysis. We have included a description of this instability in the methods section of the paper.

Results, line 387: "thechemical"

Typo corrected

Results, line 430: "fonthas"

Corrected

Results, line 438-440: Wording of this sentence is confusing. Consider rewording to improve clarity.

Rephrased for clarity as follows:

"A methyltransferase, highly expressed in *S. divinorum* trichomes, was reported in the PhD thesis of L. Kutrzeba (University of Mississippi). This enzyme showed activity in vitro in biochemical assays using radiolabeled SAM, divinorin A, and hardwickiic acid."

Discussion, lines 618-619: Are these results based on data collected by the authors of this paper or the literature? Please provide supporting information or reference.

Reference was added (PhD Thesis X. Chen)

If other genomes versions are less complete, what does that mean for comparative genomics?

Having less complete genome versions for comparative genomics has serious implications for accuracy and the types of analyses that can be performed. Incomplete assemblies result in missing or fragmented data, which can lead to misinterpretations of evolutionary relationships, lost genes, and biased conclusions. An incomplete genome may not contain all of the necessary genes for accurate phylogenetic analysis. This can lead to incorrect or unstable phylogenetic trees that misrepresent evolutionary history, especially in species with high genetic variability. Incomplete genomes are particularly problematic when trying to identify new genes, especially genes which are unique to a

species or lineage. In the end, using a less complete genome as a reference can introduce systemic biases into downstream analyses. For example, mapping sequencing reads from other individuals to an incomplete reference can cause reads to be improperly aligned or left unmapped,

label subgenera in Fig 3A

Subgenera were added.

The statement "activity of annonene synthase, in both *S. divinorum* and *S. splendens*, has evolved by recruitment of genes encoding cytochrome P450 enzymes originally with ferruginol synthase activity and neofunctionalization towards annonene synthase activity" needs to be supported with a citation or Fig. 5 needs to be updated to show which tips correspond to enzymes with FS activity.

The two previous papers showing the cytochrome P450 enzymes from *Salvia splendens* with ferruginol synthase activity (Li et al Molecular Plant 2023) and the neo-functionalization of cytochrome P450 enzymes belonging to CYP76AH subfamily to annonene synthase (Lin et al Plant Communication 2025) have been added. The cytochrome P450 enzymes with ferruginol synthase activity are highlighted with orange font in Fig. 5.

Regarding the statement "Our data suggest that in *N. benthamiana*, SdHDAS (CYP728D26) can synthesise some divinorin A and its isomer" perhaps I am missing something, but results from heterologous expression of CYP728D26 should be shown, perhaps as a supplemental figure. Or perhaps this information is coming from the previous study alluded to on line 331? Overall, there seems to be some confusion or inconsistency regarding the various P450s. Please update the manuscript to make it absolutely clear which groups did which work and when, as well as which enzymes have which activities, and present all relevant data.

The paragraph has been adjusted to address the concerns of reviewer - all the experiments in *N. benthamiana* have been repeated with at least 3 independent replicates

A number of paragraphs in the discussion are redundant with the results section and thus do relatively little to enrich the manuscript. I suggest that the discussion either be condensed, merged with the results section to create a combined section with less redundancy, or be expanded to touch on, for example, an analysis of comparative genomics as applied pathway discovery - what have such approaches, like that shown here, in combination with those preceding it, taught us overall? What are some new directions in which we could go using comparative genomics to study pathways at scale?

The discussion has been re-written to avoid repetition, providing a more concise discussion.

We thank the reviewer for this suggestion, which improves the interpretation of Figure 4. We have added the following text (lines 226 – 233).

While class II clerodane synthases – like SdKPS- can catalyze the initial cyclization and methyl migrations of GGPP to form kolavenyl diphosphate, class I activity is essential for subsequent phosphate lysis and formation of kolavenol, a critical precursor for downstream oxidations leading to furanoclerodanes like salvinorin A. The absence of conserved syntenic ortholog class I clerodane synthases genes in *Scutellaria*²⁵ and *Callicarpa* species, is in full agreement with the conclusion that there has been rapid evolution of clerodane biosynthetic pathways by convergence and that the repeated evolution a of class II diterpene synthase activity alone is insufficient for clerodane diterpene formation in the Lamiaceae.

We appreciate the reviewer's positive feedback on Figure 5 and the suggestion to explicitly link neofunctionalization to the WGD α event. We have incorporated the following text (lines 266 -276)

Since the cytochrome P450 enzymes encoded by the CYP76AH36 gene(s) show some annonene synthase activity while the enzymes encoded by the CYP76AH87 genes do not ²⁸, it can be hypothesized that SdANS and SspANS evolved from a copy of an CYP76AH36 ancestral gene. The neofunctionalization of CYP76AH enzymes in American *Salvia* species - from ancestral ferruginol synthase to annonene synthase activity - was likely facilitated by the whole-genome duplication event (WGD α) shared by these neotropical *Salvia* species. Following duplication, one copy may have retained the original ferruginol synthase function, associated with abietane diterpenoid biosynthesis, while the other acquired annonene synthase activity, enabling the formation of the furan ring in clerodane precursors such as those leading to salvinorin A ²⁸. This divergence underscores how WGD events can drive the evolution of specialized metabolic pathways in plants.